# Cilia-driven surface currents characterize specific cnidarian groups and lifecycle stages
Theres Koch [1,2,8], Karina Araslanova [1,8], Thibault Bouderlique[1,8], Anton Fetisov[1], Oliver Link[2], Alexander Klimovich [3], Milena Mičić [4], Klara Mičić [4], Johan Boström[1], Pedro Frade[5], Daniel Abed-Navandi [2,6] & Igor Adameyko [1,7] ✉

In sessile animals, body surface-associated water currents are essential for integrating feeding, cleaning, and boosting metabolic exchange across colonies or communities of solitary individuals. The evolutionary origins of surface currents and their distribution among cnidarians remain poorly understood. Here, we investigated directional surface currents by tracking moving fluorescent beads on live specimens. We show that surface-associated flows are widespread among cnidarians, including anthozoans, scyphozoans, and cubozoans, but vary in complexity. The structural organization of these currents, as well as flow regimes, correlates with animal size, coloniality, and feeding strategy, highlighting their evolutionary significance across diverse lifestyles and morphologies. Notably, we observed a consistent absence of cilia-driven surface flows in octocorals, hydrozoans, and staurozoans. Moreover, surface flow was also stage-dependent, being absent in medusae but present in polyps of the same species. This suggests that the muscle-mediated motility of a cnidarian medusae might reduce the necessity for surface-mediated hydrodynamic control in cnidarians. Overall, the patchy distribution of cilia-driven surface currents implies repeated evolutionary gains and losses under selective pressure in multiple systematic groups.

With an estimated species count ranging from 7200 to 13,000, cnidarians encompass a rich variety of lifeforms, including reef-building corals, jellyfish, sea anemones and many others[1,2]. The remarkable morphological diversity observed within cnidarians can be mainly attributed to the presence of two distinct body plans: the sessile polyp, which attaches itself to the substrate at the seabed, and the free-swimming, umbrella-shaped medusa[3].

Certain cnidarian groups have evolved key traits like modular or colonial growth, crystalline skeleton formation, and symbiotic relationships with photosynthetic dinoflagellates of the family Symbiodiniaceae, known as zooxanthellae, enabling them to thrive in diverse marine environments[4,5]. Prominent cnidarians such as hard corals (scleractinians) are recognized as ecosystem engineers due to their capacity to build robust, calcareous skeletons and construct three-dimensional structures of coral reefs[6,7].

As many cnidarians are sessile organisms, they are commonly perceived to be dependent on their surrounding environments[8]. Consequently, the immobile colonial cnidarians must seek ways to support and care for themselves and integrate neighboring polyps[9]. Quite remarkably, the physiological integration of scleractinian coral colonies is achieved in a multitude of original ways, starting from the active food-sharing movements and physical interactions between individual polyps to epidermal surface currents generated by cilia[10,11]. The cilia covering the surface of corals are capable of coordinated metachronic movements[12,13] leading to both vertical and horizontal water flows next to the body surface. These flows perform a number of essential functions. For instance, they disrupt the viscous hydromechanical boundary layer, which exists in a low Reynolds number environment, and facilitate the exchange of gases and metabolic products near the coral surface: the vertical vortices could increase the effective diffusion of nutrients and oxygen by up to 400% according to Shapiro et al.[14].

The studies of the role and structure of horizontal surface-associated currents can be traced back to the early 20th century. In 1928–29, Sir Charles

[1]Department of Neuroimmunology, Center for Brain Research, Medical University Vienna, Vienna, Austria. [2]Department of Functional and Evolutionary Ecology, Faculty of Life Sciences, University of Vienna, Vienna, Austria. [3]Zoological Institute, Christian-Albrechts University of Kiel, Kiel, Germany. [4]Aquarium Pula, Pula, Croatia. [5]Natural History Museum of Vienna, Vienna, Austria. [6]Haus-des-Meeres, Vienna, Austria. [7]Department of Physiology and Pharmacology, Karolinska Institutet, Stockholm, Sweden. [8]These authors contributed equally: Theres Koch, Karina Araslanova, Thibault Bouderlique.
✉e-mail: igor.adameyko@meduniwien.ac.at

Maurice Yonge, an English marine biologist, documented the presence of the horizontal surface currents in many scleractinian species inhabiting the Great Barrier Reef [15,16]. His observations revealed that these currents primarily serve cleansing purposes and occasionally aid in conveying food to the organism's mouth. Half a century later, Lewis and Price reported the macroscale overview of horizontal surface flows in 35 species of Atlantic and Pacific corals, reporting the general variation of the flow towards and away from the mouth or caenosarc, or tentacles in different species [17].

Despite their foundational contributions, these earlier studies did not address the challenges of modeling coral surface hydrodynamics in species with hundreds and thousands of tiny polyps due to the low spatial resolution, as well as inherent diversity, complexity, and anisotropy of water and mucus trajectories shaped by cilia-driven flows at a few millimeters scale. More recently, Bouderlique and Petersen with coauthors explored the species-specific diversity and complexity of surface hydrodynamics at both macro- and microscale in 14 scleractinian coral species [10]. The study revealed that coral surface currents exhibit remarkable structural complexity, forming intricate, meandering networks. They connect distant polyps through dynamic belts of moving water or mucus, which play a crucial role in diverse biological functions, including feeding strategies, cleaning, mucus transport, and even the resistance to tide-related air exposure. The currents hierarchically organize surface areas into anisotropically connected functional units, comprising multiple polyps. It turned out that individual polyps employ ciliary horizontal currents to capture suspended particles from shared territories and direct food toward their mouths, balancing sharing and feeding efficiency. These discoveries redefined the coral surface as a highly anisotropic, hierarchical or modular city-like structure, where unidirectional highways facilitate efficient nutrient distribution, debris removal, and protection from invading organisms. The architectural organization of such flow highways and their hierarchical tessellation emerged during colony growth, reflecting species-specific developmental patterns encoded in the species genomes [10].

The numerous benefits of surface currents observed in corals naturally lead to the question of whether similar currents exist in other cnidarian groups, ranging from solitary to colonial forms. What patterns might emerge on the surface of a jellyfish bell or along the tentacles of a hydrozoan polyp?

In this study, we aimed to investigate the presence and complexity of cilia-driven surface currents across diverse cnidarian lineages beyond highly colonial scleractinian corals, providing insights into evolutionary origins, conservation, and fundamental biological functions of horizontal surface flows in a multitude of cnidarian species.

To introduce the reader to the phylogenetic resolution in our experiments, we would like to briefly reiterate the basic aspects of cnidarian systematics. Cnidarians are divided into three major clades, each with its own distinctive morphological features [18,19]. The first of these clades is Anthozoa, which is the most species-rich class within Cnidaria. It encompasses Hexacorallia, a group that includes organisms such as scleractinians (hard corals), corallimorpharians, sea anemones, zoantharians, and antipatharians (black corals). Anthozoans also include soft corals belonging to Octocorallia (e.g., Malacalcyonacea and Scleralcyonacea), and tube anemone Ceriantharia. Medusozoa constitutes the second clade within Cnidaria. Medusozoa species undergo a life cycle alternating between sessile polyps and free-swimming medusoids. This group comprises species such as Hydrozoans (Hydroid polyps, Siphonophores, and Hydromedusae), as well as Scyphozoa (true, motile jellyfish and polyps). In addition, Cubozoa (box jellyfish) and Staurozoa (benthic stalked jellyfish) also fall within the Medusozoa clade. Lastly, the third clade, Myxozoa, is composed primarily of endoparasitic cnidarians [20,21]. Therefore, here we aimed to explore the presence and the structure of surface currents in the representatives of all systematic groups mentioned above, with the exception of Myxozoa.

Our findings presented here revealed that surface currents of varying complexity are widespread across major cnidarian groups. At the same time, notable exceptions challenged our early assumptions about coloniality

driving the necessity of polyp integration via the surface currents. Our final conclusions rather supported the orthogonal logic based on that the muscle-driven motility and recurrent contractions/pulsations anti-correlated with the presence of surface flows, leading us to think that the surface currents evolve where the motility is reduced and the sessile features, such as calcified skeletons, are maximized. Overall, our results demonstrate that horizontal surface currents represent an ancient and evolutionarily pervasive trait, offering functional advantages across cnidarians, regardless of their phylogenetic position or colonial organization.

## Results

To answer if representative species from phylogenetically diverse cnidarian groups generate complex directional surface currents, we tracked the surface hydrodynamics in live cnidarians using fluorescent trackers. We opted for fluorescent beads with the diameter of 100 micrometers or smaller, as they allowed the massive parallel tracking with sufficient spatial precision and in real time by using video recordings. The beads were introduced into the surface water or mucus layer, enabling us to map the directional and spatial patterns of flow across the cnidarian body surface (Supplementary Movie 1). The beads are non-toxic and inert, and they do not dissolve and disappear as low molecular weight dyes, also ensuring that they do not interfere with the natural physiological processes of the investigated cnidarian species. We took advantage of the particle-tracking module of IMARIS software to extract the coordinates of beads across consecutive frames and to build trajectories for every moving bead [10]. In this study, we relied on recordings of at least three or more individuals of every explored species. Often, the same individuals were recorded or observed multiple times to ensure the consistency of data prior to the analysis stage. Of note, the majority of specimens examined in this study were successfully reared and maintained in public aquaria (Haus des Meeres in Vienna and Pula Aquarium in Pula). Imaging was conducted within short periods of time and under conditions similar to routine husbandry parameters, in order to preserve animal health and maintain their native surface hydrodynamics.

By implementing further analysis of numerous collected bead trajectories, we managed to calculate parameters such as velocity, directionality, acceleration/deceleration, sinuosity, the number of turns, and overall connectivity of cilia-driven flows (see Fig. 1a for methodological aspects). This provided insights into the spatial and temporal organization of surface currents in cnidarian species generating them.

Our null hypothesis was that only highly colonial scleractinians, hydrocorals and octocorals will demonstrate the presence of complex surface currents, as those are likely evolved to respond to the challenges of coloniality and needs to integrate hundreds and thousands of individual polyps. We assumed that solitary scleractinians or scleractinians with only few large polyps comprising the colony will not show the presence of cilia-driven surface currents. Therefore, we started the bead tracing experiments with a solitary coral *Fungia sp.* and *Tubastrea coccinea*, which is made of few large polyps, or *Cladocora caespitosa*, where several large polyps are strongly separated by the common corallite (Fig. 1b, also please see Supplementary Fig. 1 for spatial arrangements of large and small polyps). To our surprise, all of them demonstrated the presence of strong surface currents with a spatial structure linking tentacles and mouths of polyps (Supplementary Fig. 2). This "stellar" type of surface currents appeared much more simple in terms of trajectories' turns and twists, being rather confined to individual large polyps (Fig. 1b), as compared to highly colonial scleractinian corals with hundreds of smaller polyps interconnected by surface flows (genera *Diploria, Agaricia, Montipora, Echinopora, Stylophora*[10]). The different spatial domains on the large oral disk transported the beads towards the mouth or away from the mouth, at the same time, thus forming the "stellar" pattern (Fig. 1b, Supplementary Fig. 2).

A similar pattern emerged in explored scleractinian relatives - corallimorpharians (genera *Ricordea, Corynactis, Pseudocorynactis*), actinarians (genera *Exaptasia, Aiptasiogeton*) and zoantharians (genera *Zoanthus*). Their oral discs and tentacles generated stellar patterns of surface-associated currents. The foot and stalk pushed beads towards the oral disk (Fig. 1b,

**Fig. 1 | Anthozoan species generate body surface-associated water currents of different complexity.** **a** A multi-step procedure used in this study: collection of specimens from their natural environment or a local public aquarium lab, followed by the application of fluorescent microbeads using a pipette, and a video recording used for the computational analyses. **b** The panels show a photograph of the organism, a visualization of bead trajectories (rendered using IMARIS), and a simplified representation of the flow pattern for the highlighted species: *Zoanthus sociatus*, *Exaiptasia diaphana*, *Cirrhipathes* cf. *spiralis*, *Corynactis viridis*, and *Fungia* sp. The recordings for studying surface currents were performed for all listed species. The modified phylogenetic tree from McFadden et al.[6], is focusing exclusively on Anthozoa and shows the relations between groups with and without surface currents. The timescale is represented as a color gradient, showing the beginning and end of the track: purple indicates the start, while orange/red represents the end of the track.

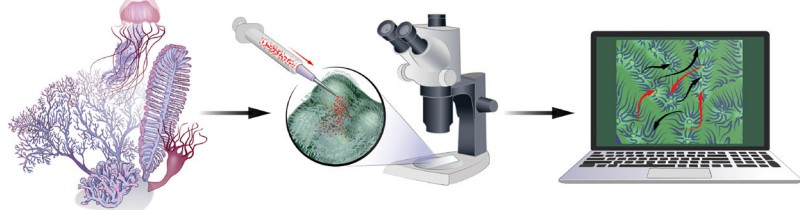

## a Experimental set-up

Cnidarian diversity → Application of fluorescent beads → Computational analysis

## b Surface currents identified in different Anthozoan groups

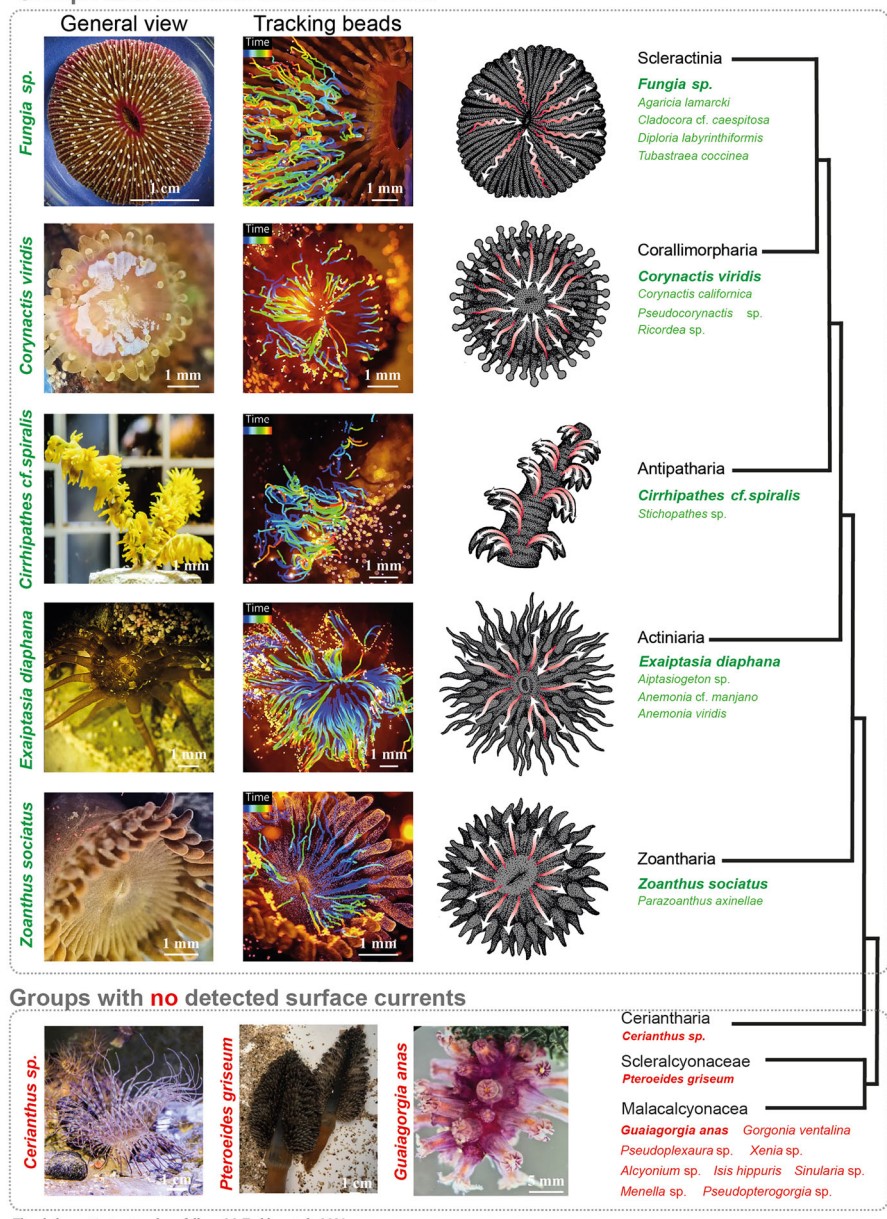

The phylogenetic tree topology follows McFadden et al., 2021

Supplementary Fig. 2). All of these animals have a typical flower-like morphology and most likely share similar ecological constrains when it comes to dealing with water flows (summarized in Supplementary Fig. 2).

However, antipatharians (genera *Cirrhipathes* and *Stichopathes*), being morphologically different, revealed a different pattern of surface-associated currents: along the main stalk of the whip-like colony, branching towards the mouth parts and tentacles of every polyp (Fig. 1b, Supplementary Fig. 2). In this case, the individual polyps were not interconnected via the surface currents, although all of them were connected hydrodynamically to the main stalk.

Despite the presence of surface currents in the aforementioned systematic groups, some animals among actinarians (genus *Anemonia*) and

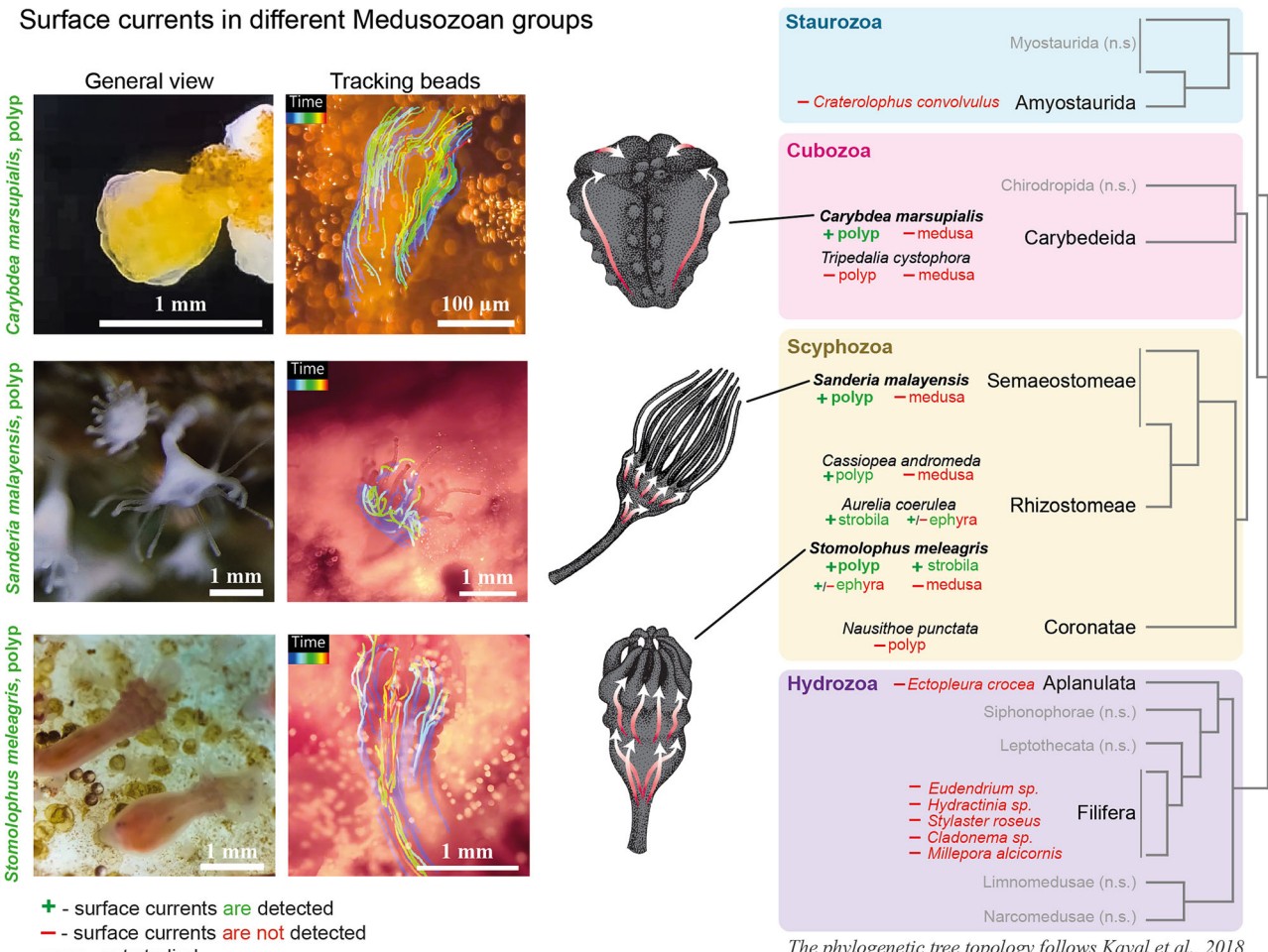

**Fig. 2 | Surface currents vary across life-cycle stages in cubozoan and scyphozoan species.** The panels show a photograph of the organism, a visualization of bead trajectories (rendered using IMARIS), and a schematic representation of the corresponding flow pattern for the highlighted species: *Carybdea marsupialis*, *Stomolophus meleagris*, and *Sanderia malayensis*. The phylogenetic tree (the phylogenetic tree topology follows Kayal et al.[18]) focuses on Medusozoa. "+" denotes species and life stages in which surface currents were detected, whereas "–" indicates life stages where no currents were observed (ephyrae generated detectable but less pronounced currents compared with polyps and strobila), groups listed in gray were not recorded for this study. Colored squares indicate the taxonomic groups. The timescale is represented as a color gradient, showing the beginning and end of the track: purple indicates the start, while orange/red represents the end of the track.

zoantharians (genus *Parazoanthus*) showed no surface-associated flows (Fig. 1b). Also, the flows were missing in Ceriantharia – another phylogenetic group within the Hexacoralia (Fig. 1b). These observations suggest that the presence or absence of surface flows might be defined by some specifics of the lifestyle, emerging or disappearing in animals within phylogenetic units.

This assumption was clearly supported by the absence of the surface currents in all explored octocorallian species from Malacalcionacea and Scleralcyonaceae (genera *Alcyonium, Gorgonia, Guaiagorgia, Isis, Menella, Xenia, Sinularia, Pseudopterogorgia, Pseudoplexaura, Pteroeides*), which are highly colonial animals with different morphologies (Fig. 1b).

The exploration of more distant phylogenetic cnidarian groups, such as Hydrozoa, also universally revealed the absence of surface-associated currents (Fig. 2). Similar to octocorals, the highly colonial hydrozoan fire corals (*Millepora sp.*) did not show surface flows despite high degree of colonial organization (for the animals without surface flow, please see the deposited raw experimental data).

Further investigation of cnidarians from Scyphozoa (genera *Nausithoe (Corinate), Sanderia, Stomolophus, Cassiopea*) and Cubozoa (genera *Tripedalia* and *Carybdea*) revealed an intriguing fact: while the polyps showed the presence of surface currents streaming from the polyp foot to the tentacles and mouth, the medusa stage showed no directional currents whatsoever (Fig. 2).

Staurozoans, or stalked jellyfish, represent a distinct cnidarian class characterized by a primarily sessile life form that resembles a polyp, although they are often regarded as evolutionarily related to medusa. Staurozoans do not generate surface currents according to our results (Fig. 3), while they show contracting movements and other types of motile behavior due to muscle activity.

To further find out, why some cnidarians or stages of their lifecycle do not generate the directional surface flows, we explored the presence or absence of cilia on their surfaces using immunohistochemistry approach. The antibodies against acetylated tubulin successfully revealed cilia covering the epithelial cells in highly colonial scleractinian *Acropra sp.* and *Echinopora sp.*, which served as a positive control with surface currents and heavily ciliated surfaces. At the same time, the surfaces of six different octocorals (genera *Guaiagorgia, Isis, Pseudoplexaura, Rumphella, Sinularia* and *Pseudopterogorgia*) showed no extending cilia at all, despite having extending cilia in their gastrovascular system (Fig. 4a, b). Thus, the surfaces of octocorals are deprived of beating cilia, in accord with some previous observations[14], which explains why they do not generate surface currents (Fig. 1).

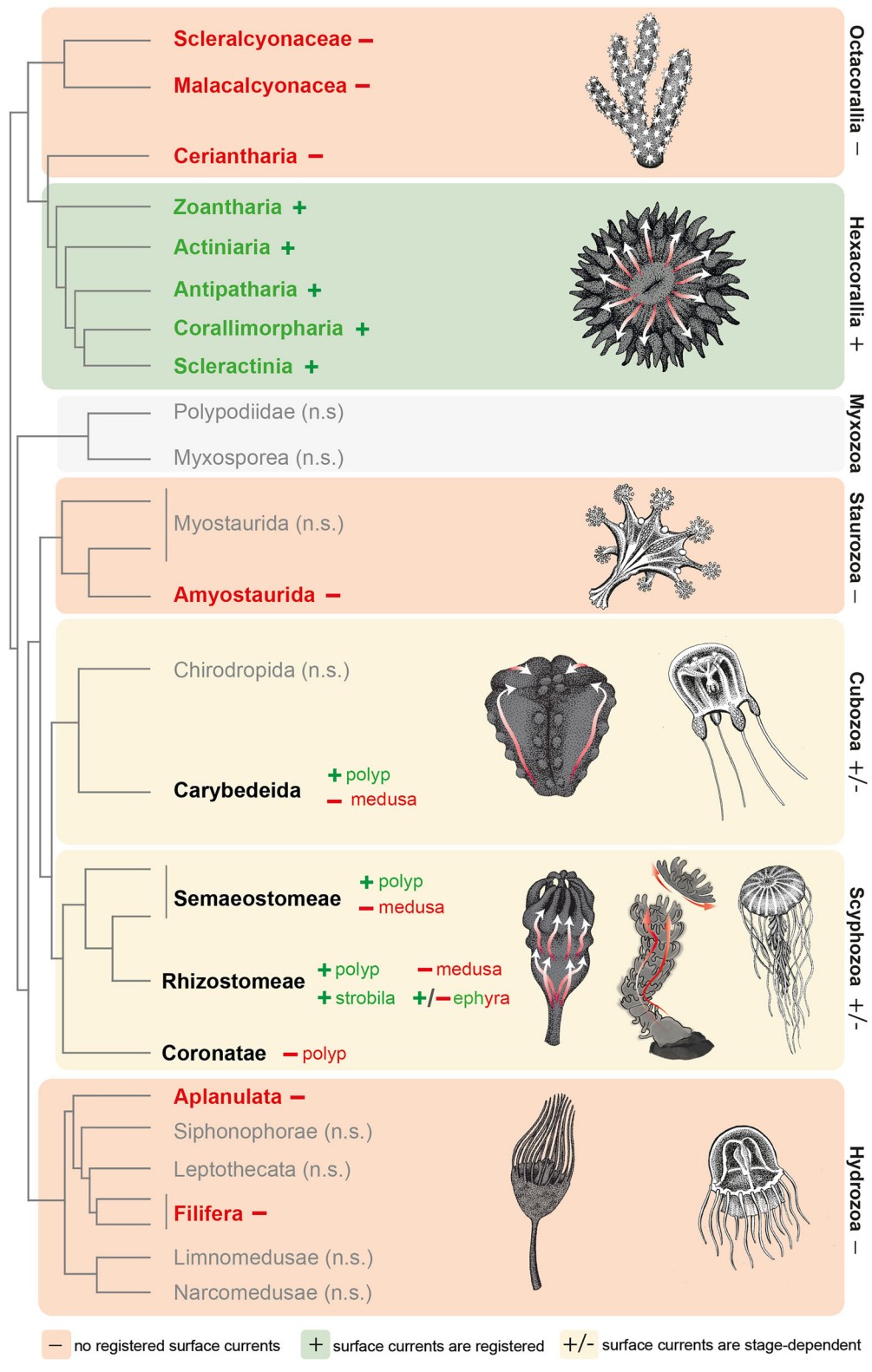

*The phylogenetic tree topology follows Kayal et al., 2018 & McFadden et al., 2021*

**Fig. 3 | Cnidarians from a wide range of systematic groups exhibit surface current generation.** This phylogenetic tree is a simplified overview of Cnidaria (the phylogenetic tree topology follows McFadden et al.[6] and Kayal et al.[18]), and provides a broad summary of surface current activity across the phylum. Green color indicates the presence of current generation, while red color denotes its absence.

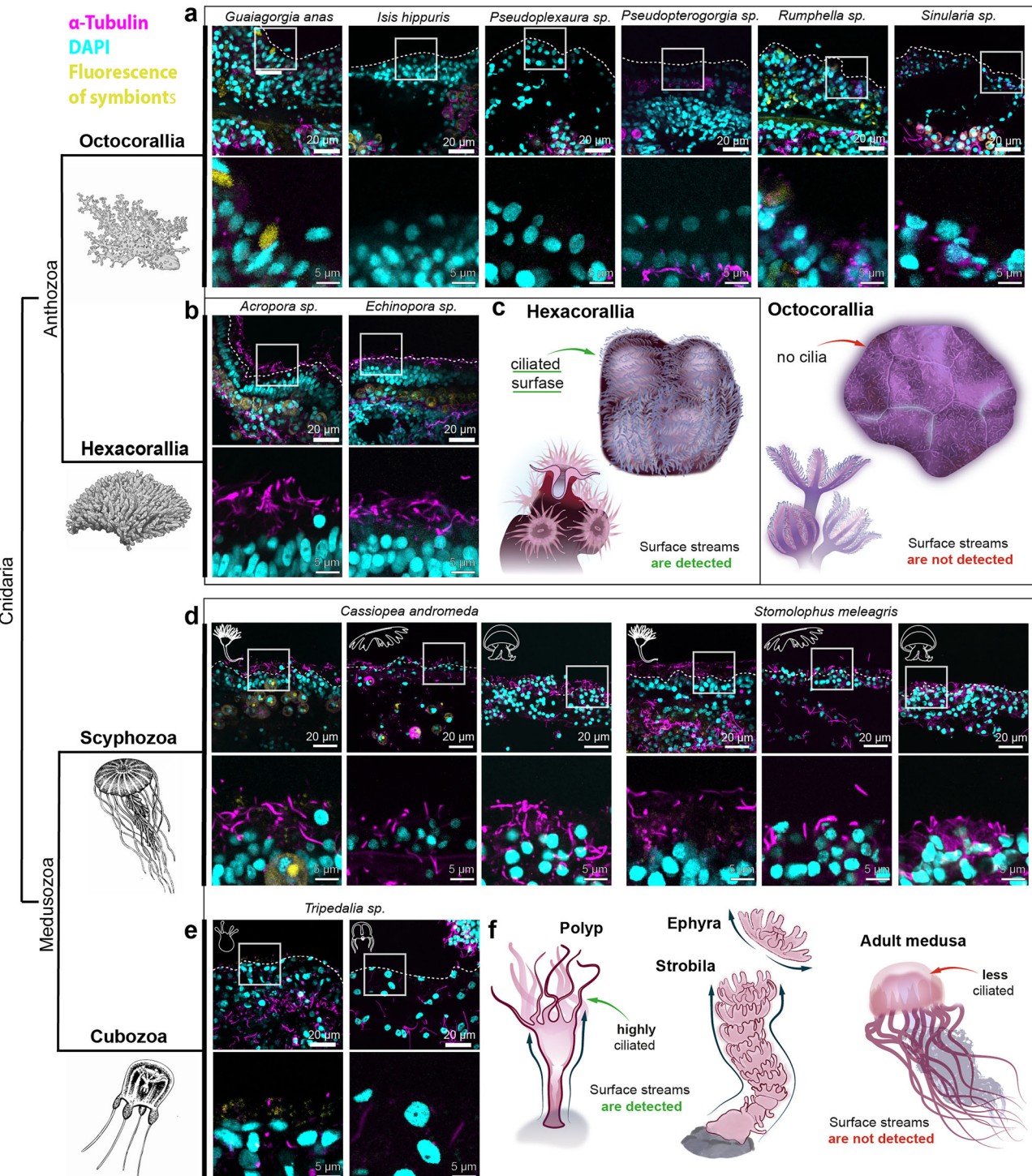

**Fig. 4 | Distribution of cilia on the surface of cnidarians correlates with presence or absence of surface-associated currents.** Cilia are stained with anti-α-Tubulin antibodies (magenta). White dashed lines indicate the outer surface of epithelia. White frames indicate magnified regions. **a** Unciliated surface of six Octocorallia species: *Guaiagorgia anas*, *Isis hippuris*, *Pseudoplexaura* sp., *Pseudopterogorgia* sp., *Rumphella* sp., *Sinularia* sp. **b** Distribution of cilia on the surface of two Hexacorallia species: *Acropora* sp., *Echinopora* sp. **c** Generalized schematic illustrating the correlation between surface ciliation and flow in Anthozoa. Hexacorallia species, which have a ciliated surface, display active surface flows, while Octocorallia species lack surface cilia and do not show detectable flows. **d** Distribution of cilia on the surface of two Scyphozoa species: *Cassiopea andromeda*, *Stomolophus meleagris*. White icons indicate the corresponding stage of a lifecycle: polyp, medusa, ephyra. **e** Distribution of cilia on the surface of Cubozoa species *Tripedalia cystophora*. White icons indicate the corresponding stage of a lifecycle: polyp, medusa. **f** Generalized schematic illustrating surface ciliation and presence of flows in different lifecycle stages of Scyphozoa. Polyps have a highly ciliated surface and show surface flows, while medusae and ephyrae stages have less pronounced surface cilia and no detectable flows.

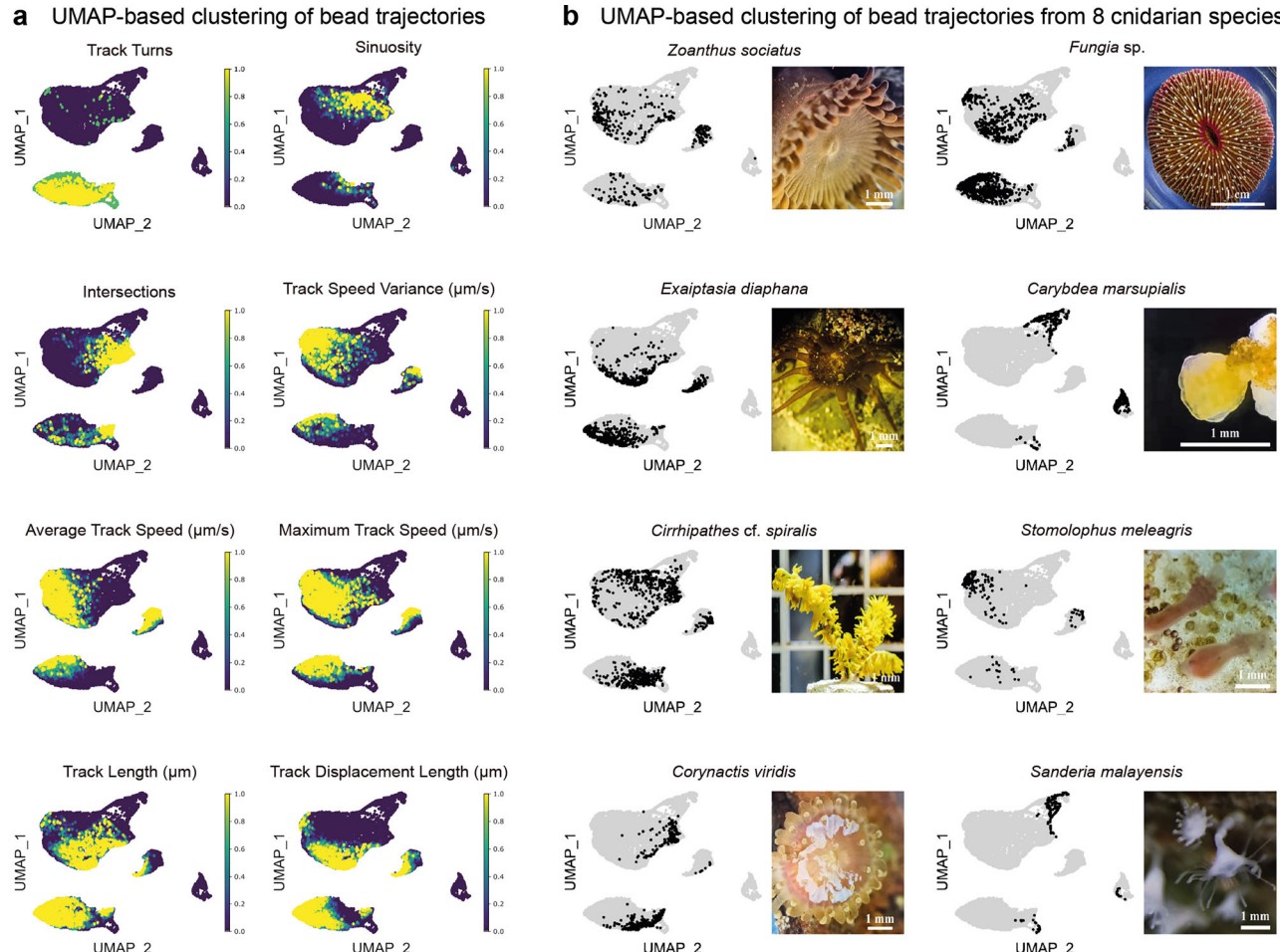

**Fig. 5 | Multivariate analysis of surface-associated flow trajectories across different cnidarian species. a** Distributions of eight quantified parameters are visualized as density gradients within a two-dimensional UMAP space. Metrics were minmax normalized to a 0–1 scale, where 0 represents the minimum observed value and 1 represents the maximum observed value for each characteristic across all tracks. **b** UMAP projections of multidimensional clustering of individual trajectories generated from eight cnidarian species. We used the tracks obtained from *Zoanthus sociatus* (*n* = 3 specimen), *Exaiptasia diaphana* (*n* = 3 specimen), *Cirrhipathes cf. spiralis* (*n* = 3 specimen), *Corynactis viridis* (*n* = 3 specimen), *Fungia sp.* (*n* = 3 specimen), *Carybdea marsupialis* (*n* = 3 specimen), *Stomolophus meleagris* (*n* = 3 specimen), and *Sanderia malayensis* (*n* = 3 specimen).

Next, we applied the same approach to investigate distribution of cilia in polyps (showing tracks), advanced ephyrae and medusa stages (no tracks) of scyphozoan *Cassiopea* and *Stomolophus*, as well as cubozoan *Tripedalia*. The immunohistochemical analysis showed the presence of cilia distributed in much lower densities in medusoid stages as compared to polyps (Fig. 4c, d, Supplementary Fig. 3), which might explain the absence of generated surface currents in medusae. Still, the transitional stages of the lifecycle - strobilae (recorded *Stomolophus meleagris* and *Aurelia coerulea*) and newborn ephyrae (recorded *Aurelia coerulea*) - revealed the presence of oriented surface currents (Supplementary Fig. 4 and Supplementary Movie 2).

Interestingly, the sensory organs of a medusoid-stage rhopalia were selectively and strongly ciliated in a manner similar to a polyp (Supplementary Fig. 5), which, hypothetically, might be required for local water turnover for performing chemosensory and mechanosensory roles[22]. This result supports that although the medusa stage might generate some degree of water and mucus dynamics, this is not sufficient to create organized horizontal flows next to the animal body surface.

In the following line of our investigation, we aimed to understand the comparative structure of bead trajectories created by different cnidarian animals to find out about possible flow regimes more systematically and in unbiased way. Trajectories of beads movement were extracted from microscopy recordings using the IMARIS tracking module, which provided

quantitative trajectory data (e.g., sinuosity, speed, turning frequency). To identify underlying patterns, these features were analyzed using Uniform Manifold Approximation and Projection (UMAP), a dimensionality reduction technique used to visualize high-dimensional data in a lower-dimensional space. After UMAP projected the high-dimensional trajectory features into a lower-dimensional space, we applied Leiden clustering algorithm to categorize distinct track types and find groups of similar data points within a network (Fig. 5a).

UMAP-based clustering reveals these fine-scale variations, enabling to detect trends that might otherwise go unnoticed (Fig. 6 and Supplementary Fig. 6). For example, UMAP projections expose intermediate flow structures, hybrid movement patterns, or rare trajectory behaviors that might be missed by human classification. Within each species, UMAP clustering highlighted that while some track types were predominant, there were also rare trajectory types that contributed to the overall dynamics. At the same time, certain species exhibited highly ordered and repetitive movement patterns, while others showed more stochastic, meandering paths (Fig. 6 – compare panels a and b). These differences may reflect alternative functional strategies in surface hydrodynamics, such as optimizing feeding efficiency, mucus transport, or self-cleaning. Yet, we are unable to ascribe precise biological functions or causal mechanisms to the observed hydrodynamic patterns. Elucidating these relationships will require targeted experimental

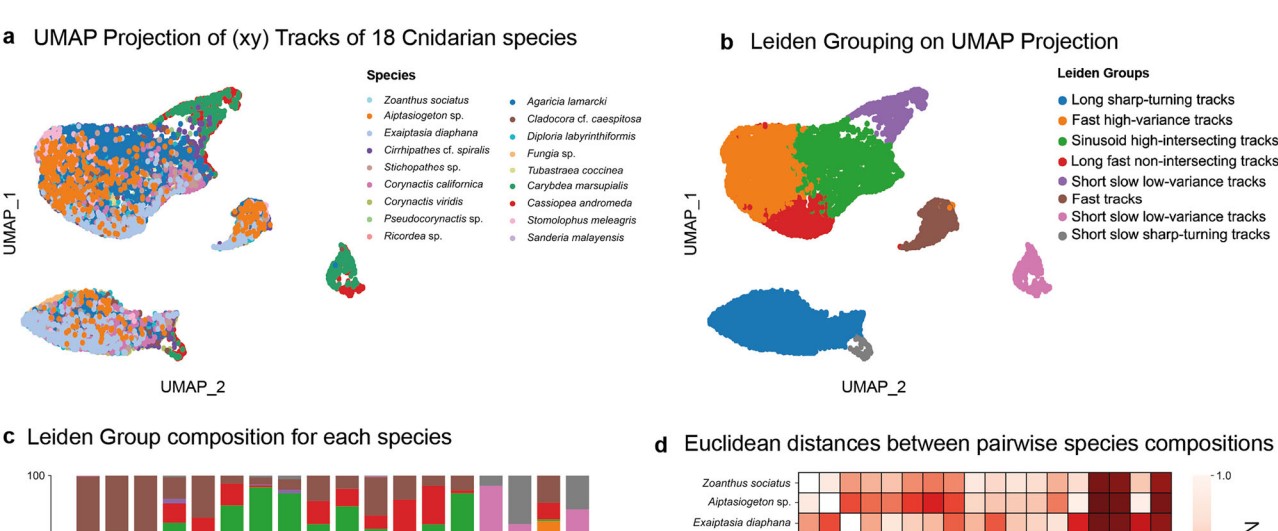

**a** UMAP Projection of (xy) Tracks of 18 Cnidarian species

**b** Leiden Grouping on UMAP Projection

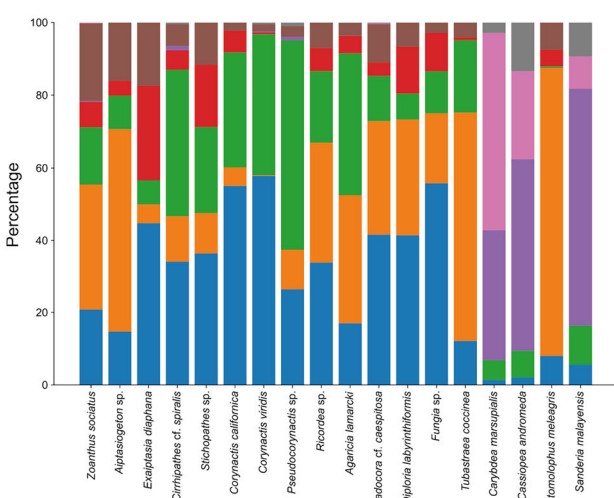

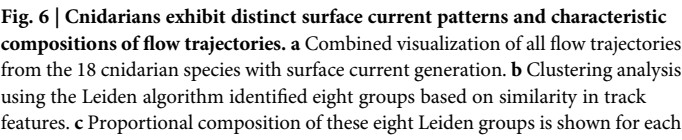

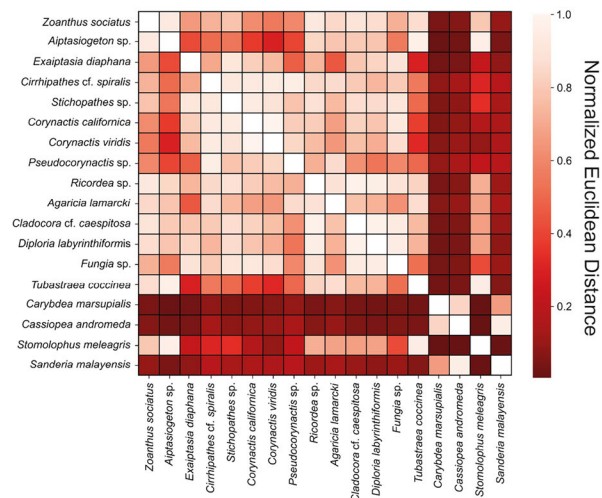

**c** Leiden Group composition for each species

**d** Euclidean distances between pairwise species compositions

**Fig. 6 | Cnidarians exhibit distinct surface current patterns and characteristic compositions of flow trajectories. a** Combined visualization of all flow trajectories from the 18 cnidarian species with surface current generation. **b** Clustering analysis using the Leiden algorithm identified eight groups based on similarity in track features. **c** Proportional composition of these eight Leiden groups is shown for each species in percentage. **d** Pairwise comparisons of group compositions, using Euclidean distances, illustrate the degree of similarity or divergence between species in terms of their trajectory group profiles. Total number of tracks = 12,512, see numbers of tracks for groups and species in Supplementary Data 1.

and theoretical work and is likely to represent a key future direction in cnidarian hydrodynamics research. Some species demonstrated unusually high diversity in their track types, suggesting either a greater range of control over ciliary movements or adaptation to variable environmental conditions.

First, let's consider the case of *Fungia sp.*, a solitary scleractinian coral that, despite its reasonably simple structure, exhibits some variation in track dynamics. The UMAP clusters indicate a mix of rather straight currents and longer, more turning and twisting flows (Fig. 6b). In fact, these two flow regimes reflect the movements of particles along the corallite ridges towards the mouth and in the opposite direction – away from the mouth (Fig. 1b, Supplementary Fig. 2). Some particles are caught in the zones, where they make turns and, therefore, change directions, which is emerging as the second type of longer turning tracks in the UMAP.

Among the other species with the higher intra-species variation of tracks, *Cirrhipathes cf. spiralis* and *Exaiptasia diaphana* stand out with specific distribution of trajectory types, as revealed by UMAP clustering and Leiden grouping (Fig. 6). The bead tracks in these species display variation in speed, directionality, and sinuosity, indicating flexible and dynamic ciliary activity. Some trajectories are highly directed and linear, while others follow chaotic, sinuous paths (Figs. 5 and 6, Supplementary Fig. 2).

Many of such species display track diversity patterns that do not necessarily correspond to their morphological complexity. For instance, *Zoanthus sociatus* shows a degree of trajectory heterogeneity, despite its relatively simple anatomy. In that specific case, UMAP clustering suggests

that this variation is primarily driven by differences in track speed and displacement length (Fig. 6, Supplementary Fig. 2).

On the other hand, polyps of *Carybdea marsupialis*, a box jellyfish, demonstrate a very distinct track profile with tightly clustered, highly uniform rather straight or a little bit curved trajectories (Fig. 6, Supplementary Fig. 2). The same pattern is observed in the scyphozoan polyps of *Sanderia malayensis*, which resemble the polyps of *Carybdea* in their "bell shape" (Supplementary Fig. 2, Fig. 6, Supplementary Fig. 5). However, the scyphozoan polyps of *Stomolophus meleagris* have an extended, more tubular design, and they generate longer, straighter and faster tracks along their surface (Fig. 2, Supplementary Fig. 2). This suggests a high degree of coordination in ciliary beating, resulting in consistent and efficient monotypical surface currents reflected in the UMAP. The tracks observed in scyphozoan and cubozoan polyps are universally going through the foot to the mouth, however, with different speed-related nuances. Thus, the species from different systematic groups yield tracks with similar properties in cases of converging body shape, all according to the computational analysis and direct observations of bead trajectories.

From the UMAP, it is clear the stereotypicity of the flow structure in scyphozoan and cubozoan polyps contrasts with the greater diversity seen in other cnidarian groups, making scyphozoan and cubozoan polyps an interesting case of streamlined fluid dynamics. Hypothetically, this can be additionally explained by the small size of the polypoid stage (0.5–5 mm, see Fig. 7), as compared to larger colonial and solitary cnidarian life forms. This

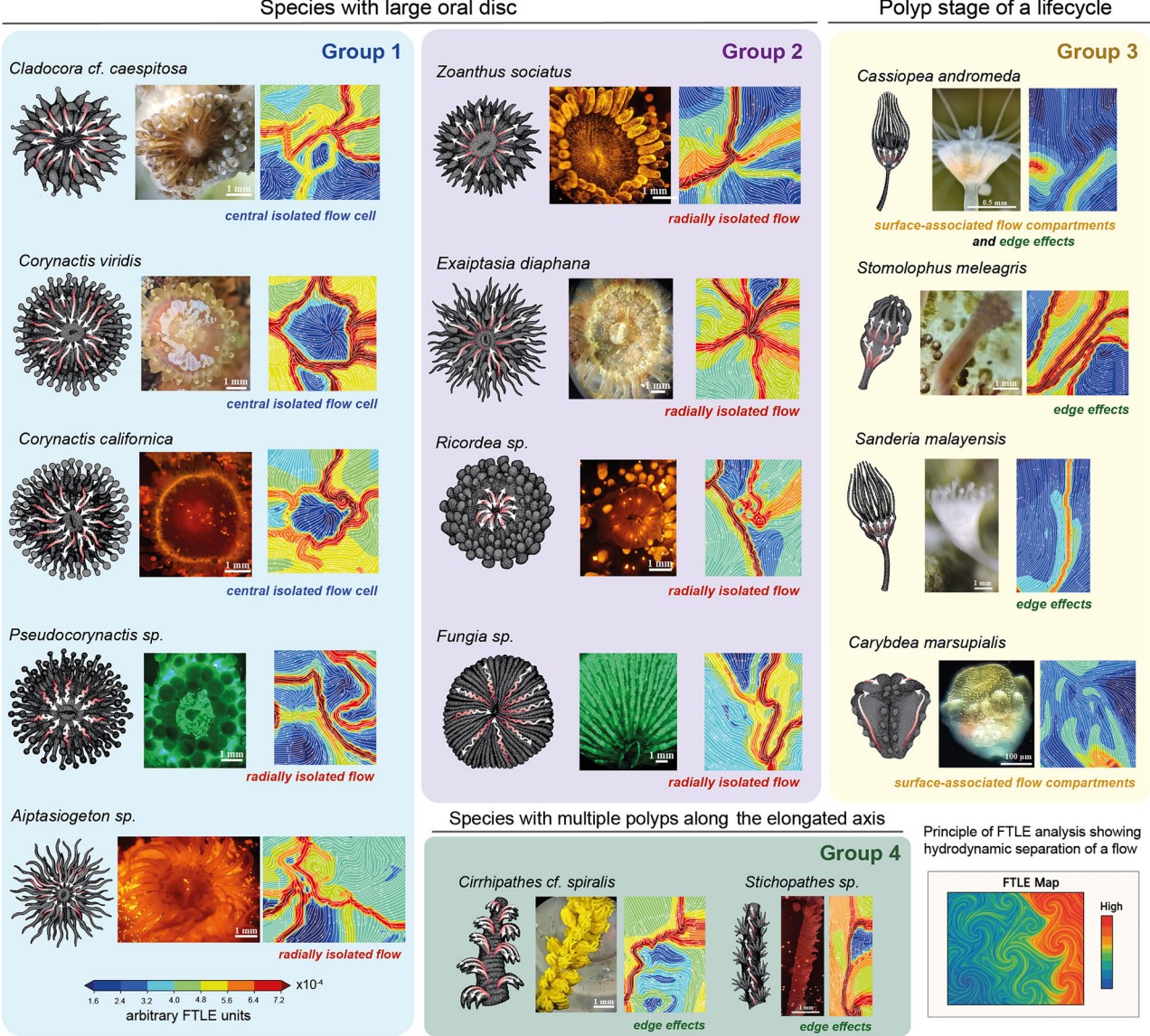

**Fig. 7 | Summary of stereotypical flow patterns and corresponding FTLE maps.** The patterns are grouped into four overarching groups: (Group 1) Currents primarily or exclusively directed toward the mouth (*Aiptasiogeton* sp., *Corynactis viridis*, *Pseudocorynactis* sp., *Corynactis californica*, *Cladocora* cf. *caespitosa*, and *Carybdea marsupialis*); (Group 2) Currents predominantly directed away from the polyp mouth (*Zoanthus sociatus*, *Exaiptasia diaphana*, *Ricordea* sp., and *Fungia* sp.); (Group 3) Longitudinal currents in polyp stages of scyphozoans (*Cassiopea* *andromeda*, *Stomolophus meleagris*, *Sanderia malayensis*) and cubozoans (*Carybdea marsupialis*); (Group 4) Currents in filament-shaped colonial species with few polyps (*Cirrhipathes* cf. *spiralis* and *Stichopathes* sp.). Note the formation of edge hydrodynamics featuring high speed in surface-associated layer and the presence of hydrodynamically isolated units in some animals (mouth areas in *Corynactis* and *Cladocora*). The color bar is applicable for all FTLE images shown.

is supported by the observation that larger (5–15 mm) species with similar "polypoid" morphology, such as *Exaiptasia diaphana* and *Zoanthus sociatus*, generate more diverse flow patterns.

Next, we applied the Finite-Time Lyapunov Exponent (FTLE) analysis to the trajectories of tracked beads served to profile complexity of transport barriers and investigate the regions of converging and diverging flow (Fig. 7). The FTLE is a mathematical tool used to analyze the complexity of fluid flows, particularly to identify coherent structures in unsteady, time-dependent hydrodynamic environments such as ocean currents, or atmospheric circulation[23]. The FTLE results revealed that transport barriers form along the tentacles of anthozoan polyps, often creating flow-isolated compartments corresponding to the central disk with mouth, suggesting a "sucking" mode of hydrodynamics in this area (see *Corynactis* species in Fig. 7). In scyphozoan and cubozoan polyps, FTLE highlighted the flow

boundaries next to the surface of the polyp, which engage the surrounding water into a systematic movement from foot towards the mouth. Overall, our computational analysis shows that non-scleractinian cnidarians do not produce the highly complex, twisting surface flow tracks seen in highly colonial reef-building corals. Track complexity tends to correlate with the presence of many small polyps, whereas the anthozoan species with few large or solitary polyps generate simpler flow patterns.

## Discussion

Inability to perform movements for escaping from predators or for changing the local environment for a better condition limits animals in opportunities and competitiveness. As response to immobility, the sessile animals have to evolve and boost other features, compensating for the loss of extensive physical motion[24]. One such approach the sessile animals take is

the active control of the hydrodynamics at their body surfaces[10]. In addition to this, sessile colonial organisms need to achieve functional integration of multiple polyps or zooids, enabling fair food sharing, joint cleaning of the colony surface and many more[10,25].

They achieve it via the coordinated beating of cilia covering surface areas, which altogether create highly complex, long and directional currents pushing seawater mixed with slime along specific trajectories spanning the surface. For instance, this capacity has been recently studied in scleractinian corals, where directional surface streams integrate thousands of individual polyps[10]. Being found in every investigated species of scleractinians, the highly complex directional surface-associated currents hint the uniqueness of scleractinians among other cnidarian groups. Also, our previous results[10] suggest a hypothesis implying that high coloniality and the integration of numerous tiny polyps promote intricate surface-associated flow dynamics. To test such uniqueness of scleractinians, we explored the presence of surface-associated currents in a broad spectrum of cnidarian groups[19] beyond scleractinian corals, including additional Hexacoralia groups (Zoantharia, Actinaria,) other more distant Anthozoans (Ceriantharia, Octocorallia, Corallimorpharia), and representatives of neighboring major systematic groups such as Hydrozoa, Scyphozoa, Staurozoa, and Cubozoa.

At the core of our methodology, we relied on a massive parallel tracking of individual tiny particles (smaller than 100 micrometers in diameter) administered to the surface of cnidarian animals, with following computational analysis of recorded tracks for extracting directionality, speed, spatial patterns and other features of particle tracks for using them in a comparative analysis framework. Our null hypothesis was based on the assumption that scleractinian corals likely achieved the biological success because of unique directional currents on their surface, which enabled better integration of their colonies. This logic was further strengthened by the fact that marine sponges – another group of sessile animals with filter-feeding lifestyle and numerous repetitive filtering units, are using surface currents for moving and sneezing mucus with trapped clogging particles[26].

Firstly, our results indicate that directional surface-associated currents are not restricted to highly colonial scleractinian corals, but are also present across multiple other cnidarian groups. In all cases, the observed surface-flow patterns were anatomically coherent and scaled with the gross body plan: comparable flows could be detected in cnidarians with simpler morphologies and a low degree of coloniality. However, in these forms the trajectories remained comparatively simple when contrasted with those generated by highly colonial scleractinians, whose surface flows and FTLE fields were tracked and mapped in detail by Bouderlique et al.[10]. To specifically assess how the degree of coloniality influences the presence and structure of surface flows, we examined *Fungia sp.*, a solitary scleractinian coral, as well as other scleractinians composed of only a few large polyps with broad oral discs. These life forms exhibited simplified surface currents that were predominantly directed either from the tips of the arms towards the mouth or in the opposite direction, and we did not detect flow patterns that integrated multiple polyps into a common regime. In contrast, highly colonial scleractinians produced trajectories with numerous turns and complex geometrical configurations, effectively partitioning the colony surface into organized feeding zones or territories that connect groups of polyps[10].

Secondly, contrary to our initial expectation that a high degree of coloniality should be associated with complex surface currents, we found no detectable surface currents in octocorals or in hydrozoan fire corals, even though these taxa often form colonies comprising hundreds to thousands of individual polyps[27,28]. Our tracking experiments in several octocoralian species and also the immunohistochemistry analysis of their cross-sections revealed the absence of extending cilia specifically on the outer surfaces, but not in the gastrovascular system, where the cilia were present at high densities. Besides, the cilia in the gastrovascular system of octocorals are known to be actively driving the internal flow[29].

The systematic absence of surface flows in different octocorallian groups with different colony morphologies is puzzling, as there must be features that enable them to achieve the same goals as in scleractinian corals without generating surface current activity. For instance, they can potentially compensate for the poor gas and metabolite exchange at the colony surface with more intense cilia-mediated flow in the gastrovascular canals and gastric cavities of the individual polyps[29,30]. Also, the hydrodynamics in the octocoral gastrovascular network might be sufficiently adapted to transport the colony-integrating signals, food components and symbionts. Similarly to octocorallian groups, Hydrozoa universally showed no detectable surface-associated currents while having internal flows in their gastrovascular system[29]. We concluded that surface streams were not strictly necessary for coordinating multiple polyps within a colony, although complex surface hydrodynamics can be beneficial in many ways. Overall, this points to the idea that scleractinian corals uniquely utilize surface hydrodynamics for the benefit of the colony, which may have favored their impressive persistence in the biosphere.

While sessile scleractinian polyps rely heavily on hydrodynamic coordination via surface-associated flows, octocoral polyps exhibit remarkable extension and motility[31,32], enabled by well-developed musculature, in stark contrast to the more rigid and sessile scleractinian corals. Additionally, octocorals lack a rigid, continuously calcified skeleton, which grants them flexibility and allows them to passively align with changing ocean currents more effectively[33]. This combination of muscle-mediated movement and the absence of a rigid skeletal structure differentiate octocorals from scleractinian corals and suggests a possible anticorrelation between motility, pulsations and the presence of cilia-mediated directional surface currents. It is conceivable that the inability of certain cnidarian groups to generate cilia-driven surface flows may have fueled the evolution of specialized neuronal circuits to coordinate rhythmic contractile activity. This, in turn, could have led to the emergence of a distinct neuronal cell type - pacemaker neurons. Supporting this idea, pacemaker neurons have been identified in several hydrozoan and scyphozoan species[34], which rely less on ciliary surface flows, whereas they appear to be absent in sessile corals, where cilia-based transport plays a more prominent role.

To test this general hypothesis about anticorrelation of motility and surface currents, we examined the medusoid stages of scyphozoan and cubozoan cnidarians, which had previously tested positive for directional surface currents during their polyp stage. Since medusae rely on rhythmic muscular contractions for propulsion, we sought to determine whether surface cilia remained functionally active at this stage. However, we detected no directional surface currents in any of the medusae studied. To explore whether this absence of currents correlated with a loss of cilia, we examined the bell surfaces of scyphozoan and cubozoan medusae and found that cilia were indeed present. However, their density and distribution were significantly reduced compared to the polyps of the same species, which were capable of generating surface currents. The highest ciliary density was observed on rhopalia, but even there, the activity and organization of cilia appeared insufficient to produce directional water or mucus flows extending towards the whole bell. Although ciliary activity at the surface is expected to enhance vertical water movement in general, thereby improving gas exchange and metabolite transport[14], the emergence of directional surface currents depends on the degree of coordination among cilia[35–37], with only sufficiently synchronized beating generating sustained and organized flow patterns.

This discrepancy between polypoid and medusoid life forms further supports the hypothesis that increased motility reduces the necessity for ciliated surface flows. A parallel no-cilia and no-current trend is evident in hydras: despite their polypoid organization, hydras exhibit recurrent, rhythmically paced muscular contractions, which are essential for their survival[38]. Inhibition of these contractions leads to their death. Consistently, our investigation revealed no surface currents in any examined marine hydrozoan species at any stage of their lifecycle. Finally, in line with this logic, we observed that staurozoans (stalked jellyfish) lack detectable surface current activity, while being capable of sporadic contractions and movements[39]. Taken together, these findings inspire a hypothesis of a fundamental evolutionary trade-off: as muscle-based motility increases, the functional role of ciliated surface currents diminishes, possibly because

muscular activity provides alternative mechanisms for maintaining environmental interactions such as feeding, waste removal, and fluid dynamics. From a phylogenetic perspective, this suggests that large cnidarian classes, such as Hydrozoa and Staurozoa, may have lacked surface currents since their initial divergence from other cnidarian lineages.

Alternatively, the presence or absence of self-generated surface currents in highly colonial cnidarians may reflect differences in sensitivity to ambient flow during early development. Many such species settle in shaded microhabitats, for example, under fully grown corals or within reef crevices, where background flow is minimal[40,41]. In such contexts, the capacity to generate localized water movement may confer a selective advantage enhancing nutrient delivery and waste removal, and thereby broadening the range of viable settlement sites. As these organisms mature and their colonies expand, they may become less dependent on external or self-generated flow for survival, relying instead on their established structural complexity and improved exposure to ambient flow[42,43]. For example, one of the modes of propagation of scleractinian corals – through fragmented nubbins or detached branches – might be rather independent of ambient flow conditions for effective resettlement, as scleractinians can effectively propagate in lower-flow, shaded microhabitats, where fallen fragments may readily initiate new colonies (Lirman, 2000). Conversely, in other colonial organisms such as octocorals, successful settlement and growth may require exposure to already well-developed oceanic currents, thereby restricting them to more hydrodynamically favorable habitats. According to our results presented here, unlike reef-building scleractinians, octocorals lack the ability to generate surface currents autonomously and instead must rely exclusively on ambient hydrodynamic forces. Also, octocorals are not fragile as compared to scleractinians and do not propagate through fragmented nubbins falling immediately next to the maternal colony[44]. Overall, these contrasting strategies underscore the role of hydrodynamic interaction in shaping the ecological niches and dispersal capacities of colonial cnidarians.

The more complex situation is observed among groups of anthozoans with lower degree of coloniality, where we have a mix of smaller systematic groups with surface cilia-driven currents and without surface cilia-driven currents, which rather supports the possibility of a secondary loss of directional surface currents, for example, in ceriantharians. Unlike corals and many other anthozoans that rely on small-scale water currents to facilitate particle capture, ceriantharians possess two distinct sets of long, highly mobile tentacles: their outer tentacles actively extend into the water column to trap plankton, while their inner tentacles transport food toward the mouth[45]. This direct tentacle-mediated feeding strategy may reduce reliance on cilia-generated flows for nutrient acquisition, which appear dominating in anthozoans with "stellar"-shaped flows connecting tips of the tentacles with oral disk and mouth. Furthermore, even the reasonably simple "stellar" pattern of flows in polypoid forms is not uniform according to our computational analysis of individual trajectories of beads, as we identified several flows regimes related to inward and outward movements of beads, as well as turbulent situations proximally to the mouth.

These insights naturally raise the question of whether directional surface currents in cnidarians represent an ancestral, monophyletic trait - suggesting that they were present in the earliest cnidarians and subsequently lost in certain lineages - or whether they evolved independently multiple times in response to similar selective pressures. Our current data do not allow us to definitively resolve this evolutionary conundrum. However, the observed variability across cnidarian groups suggests that ciliated body surfaces may have repeatedly emerged and disappeared throughout cnidarian evolution, depending on ecological and functional constraints. Support for this hypothesis comes from life cycle alternation within the same species. In ceriantharians and octocorals, for instance, planula larvae are ciliated, facilitating dispersal and substrate selection, while adult polyps lack ciliated body surfaces and rely on other mechanisms for interaction with their environment. The scyphozoan medusa stage does not generate directional surface currents, whereas the polyps and strobilae do. Similarly, in adult octocorals and hydroid polyps, directional fluid movements are generated internally within the gastrovascular system by oriented ciliary

beating, even though the external body surface lacks this ability[29]. This indicates that cilia-driven surface currents can be retained in specific physiological contexts while being lost in others, depending on its adaptive necessity, which does not depend on the general capacity of an animal to generate ciliated epithelia.

Future research should expand the taxonomic profiling of cnidarians, with denser sampling within narrower clades and comparative analyses across major lineages. This will be essential for determining whether ciliated external surfaces and horizontal flows share a single evolutionary origin or have arisen multiple times through convergent evolution. Important open questions also concern the physiological control of these surface currents – both nervous system-dependent and independent – as well as their modulation by diurnal cycles and feeding dynamics, all of which lie beyond the scope of this study. A further critical challenge is to elucidate the developmental and molecular mechanisms that establish collective ciliary orientation across polarized epithelial fields[46]. Although directional coordination of ciliary beating is a fundamental property of many tissues, the underlying regulatory pathways remain poorly understood, not only in cnidarians but even in vertebrates, including humans[37,47]. Addressing these mechanisms in cnidarians – among the earliest animals to evolve directional cilia-driven surface flows – may reveal general principles of ciliary coordination and illuminate the deep evolutionary origins of cilia-mediated transport across the animal kingdom.

## Methods

To answer if representative species from a wide plethora of phylogenetically diverse cnidarian groups generate complex directional surface currents, we tracked the surface hydrodynamics in live cnidarians using tiny fluorescent trackers. The experimental set-up involved a standardized specimen handling, bead application, and high-resolution imaging and computational analysis to ensure comparability across species. A detailed list of key materials we used in this study is provided in Supplementary Data 2.

### Animals

With the exceptions described below, all animals were reared, maintained, and propagated ex situ at Haus-des-Meeres (HdM), a public marine aquarium in Vienna, Austria. Specimens were kept in closed-circuit seawater systems under stable environmental conditions and standardized husbandry protocols, and many colonies were sustained over prolonged periods with regular growth and propagation.

The illumination of the tanks was provided for 12 h daily at ~100–140 μmol photons m$^{-2}$ s$^{-1}$, as measured with a Quantum Meter QMSS (Apogee Inc., USA). Aquaria were filled with seawater prepared from an artificial salt mix (Reef Salt, Aqua Forest Inc., Poland) dissolved in drinking water and adjusted to a salinity of 35 PSU. Water movement over the corals was generated by flow pumps (Turbelle Stream, Tunze Inc., Germany), producing velocities of 5–10 cm s$^{-1}$. Temperature was regulated by a thermostatically controlled heater (Jaeger/Eheim Inc., Germany), and the inorganic seawater composition was routinely assessed by ICP-OES (Oceamo Inc., Austria).

Routine monitoring of behavior, polyp extension, and tissue appearance indicated that the animals remained in good health throughout the study. Feeding regimes and water temperature settings for each species were maintained according to the husbandry conditions used at the institutions from which they were obtained. Cubozoa samples were obtained from the culture at Vienna Zoo, Vienna, Austria. Pennatula animals were maintained and recorded at Pula Aquarium, Pula, Croatia. Hydrozoan, Staurozoan and cold water octocoralian specimens were recorded at Helgoland Biological Station, Helgoland, Germany, they were used for the undergraduate student marine biology course (species identification and biodiversity study). No excessive disturbing manipulations were done prior to recordings. The remaining three species (*Millepora alcicornis*, *Stylaster roseus*, and *Stichopathes sp.*) were examined during a two-week research excursion at the CARMABI research station in Curaçao, Southern Caribbean Sea, in September 2022 (the animals were previously collected from 10 meters depth

and hosted in local tanks). The specimens are part of an exchange between the CARMABI Institute (CITES #AN001) and the Natural History Museum Vienna (Naturhistorisches Museum Wien, CITES #AT017). All material was collected in Curaçao (former Netherlands Antilles) under collection permits issued to CARMABI by the Government of Curaçao (permit CARMABI 2019/021824).

## Fluorescent microbead preparation

To visualize surface-associated flows, we opted for fluorescent microbeads of two size classes (1–5 μm and 63–75 μm in diameter) and densities of material (1.3 g cm$^{-3}$ and 1.090 g cm$^{-3}$ respectively) as they allowed massive parallel tracking with sufficient spatial precision. Beads were coated with Tween 80 for 12 hours, washed five times with PBS, and centrifuged at $1000 \times g$. For each species, 2–3 mL of prepared beads were resuspended in 7–8 mL of unfiltered seawater from the respective species' environment and centrifuged at $2000 \times g$ for 10 min. This washing step was repeated three times. To ensure the beads remained suspended within the surface layer, salinity was adjusted to achieve near-neutral buoyancy, resulting in sedimentation rates below 1 cm min$^{-1}$. This preparation allowed the beads to trace near-surface currents without floating away or settling prematurely. For species which were particularly small in size - such as *Carybdea marsupialis*, polyps of *Cassiopea andromeda*, *Sanderia malayensis*, and hydrozoans from the genera *Cladonema*, *Eudendrium*, and *Hydractinia* - we exclusively used the smaller beads (1–5 μm).

## Specimen preparation and laboratory experiments

Immediately after collection, specimens or colonies were transferred to the laboratory and placed in glass containers (20 × 10 cm) filled with seawater from their original environment (either in the aquarium or their natural habitat). Each specimen was covered with 1–2 cm of filtered seawater and allowed to acclimate for one to four hours in an artificial light-free environment with provided water current and conditions similar to maintenance environment. Typically, at least three individuals or more were examined per species with very few exceptions: for *Millepora alcicornis*, *Agaricia lamarcki*, and *Gorgonia ventalina*, only two colonies were available, while live *Cerianthus* sp. was represented by a single specimen. Also, for the recordings of Scyphozoan strobilae, we used *two Stomolophus meleagris* and two *Aurelia coerulea* strobilae, due to their scarcity.

Recordings were performed under a white light or with assistance of fluorescent stereomicroscope (Leica M165 FC) equipped with a CoolLED light source (light intensity 200 μmol m$^{-2}$ s$^{-1}$ (photosynthetic photon flux)) and a Samsung X23 camera (24 frames per second). Approximately 100–200 μL of the bead suspension was gently applied using a micropipette, into the surface water or mucus layer of the species, with beads released 1–5 mm above the specimen or colonies. Beads reliably adhered to the animal's surface due to their density and low vertical drift, which enabled us to map the directional and spatial patters of flow dynamics. Imaging began once the water had settled and continued until bead movement ceased or the surface was cleared, with each recording lasting between three and five minutes. Polyp size was determined by comparing it to the known size of beads in the same frame.

## Track creation and analysis

Video processing followed the procedure described by Bouderlique et al.[10]. Briefly, frame extraction was performed using ffmpeg, saving every second frame as a TIFF image. Once the frame rate was determined, every second frame was extracted to optimize computational efficiency. The frames of the original recordings were converted to time codes in seconds (see here: https://dataverse.harvard.edu/dataset.xhtml?persistentId=doi:10.7910/DVN/IMXADV). This subsampling was accounted for in the subsequent analysis of track speeds. The scripts were placed in the directory containing the videos and adapted to match the appropriate file extension. The resulting image series was reassembled and converted into IMARIS format using the IMARIS files using the IMARIS File Converter tool.

Next, we took advantage of the semi-automated particle-tracking module of the IMARIS software. For each recording, raw tracking data were exported as. csv files directly from IMARIS. Based on the bead trajectories, we managed to calculate key parameters describing the tracks, including track length, speed, number of track turns and curvature. Track curvature was expressed using sinuosity, a dimensionless measure of the straightness or curvature of a path[48].

Sinuosity was calculated as the ratio of the total track length to the straight-line distance between the track's start and end points, with values greater than 1 indicating a curved path. To analyze local variations in sinuosity along each track, we applied a sliding window approach in which a 20-point window moved one point at a time along the track. The window size depends on the scale of the animal and is selected so that each particle is tracked every second frame. Therefore, the distance varies according to the video's frame rate and the specimen's size. For each window position, the sinuosity of the corresponding 20-point segment was calculated. At each point along the track, the highest sinuosity value from all overlapping windows was retained and assigned to that position.

Track turns, we estimated by identifying segments where the sinuosity exceeded a threshold of 1.2 (the value at which random sinuosity stops exceeding the observed sinuosity[49], with each such segment counted as a turn. Self-intersections, mean, maximum, and variance of speed as well as total track length and track displacement length (i.e. the spatial distance between the first track point and the last track point) were also determined.

To classify the tracks, we applied the Scanpy Python tool to a dataset comprising z-scores of the number of turns, sinuosity, average track speed (μm sec$^{-1}$), maximum speed (μm sec$^{-1}$), speed variance (μm sec$^{-1}$), the number of intersections, track length (μm), and track displacement length (μm). By applying dimensionality reduction via Uniform Manifold Approximation and Projection (UMAP), we projected these multi-dimensional track features into a lower-dimensional space where clustering could reveal distinct categories of movement patterns. UMAP is a dimensionality reduction technique used to visualize high-dimensional data in a lower-dimensional space, typically two or three dimensions[50]. In the context of this study, UMAP was applied to bead trajectory data to reveal patterns in surface currents across different cnidarian species. Unlike other dimensionality reduction techniques, such as Principal component analysis (PCA) (which assumes linear relationships) or t-SNE (which focuses more on preserving local neighborhoods but can distort global relationships), UMAP balances local and global structure, making it ideal for analyzing complex trajectory data with underlying patterns.

After UMAP projected the high-dimensional trajectory features into a lower-dimensional space, a clustering algorithm was applied to categorize distinct track types. A nearest neighbors graph was constructed from the extracted features using correlation distance as the similarity metric. Based on this graph, Leiden clustering was performed (resolution = 0.05), along with UMAP dimensionality reduction (min_dist = 0.2). Leiden clustering is particularly well-suited for this type of data because it detects natural groupings in the projected space, meaning that trajectories with similar movement properties (e.g., sinuosity, speed, directionality) form distinct clusters. Also, it works well with network-based representations, aligning with the graph-based nature of UMAP. By combining UMAP with Leiden clustering[51], we were able to identify and visualize distinct track categories, revealing both interspecies and intra-species variation in surface currents. Together, this provided a high-resolution, data-driven classification of bead trajectories, uncovering patterns that would be difficult to detect with traditional manual categorization. Unlike manual classification, which relies on subjective interpretation, UMAP-based clustering groups high numbers of trajectories based on quantitative features, ensuring consistency and reproducibility.

Differences in feature values between clusters were assessed using t-tests. The proportion of each species within individual clusters, compared to all other species, was evaluated using a permutation-based approach. A null distribution was generated over 1000 iterations by randomly dividing all tracks into two groups matching the original group sizes. For each

iteration, the proportional difference between the two groups was calculated for each cluster. These values were then compared to the originally observed proportional difference for each cluster. The *p*-value was calculated by dividing the number of randomly generated proportional differences that exceeded the observed value by the total number of permutations.

FTLE analysis: The trajectories of tracked beads served as the raw input for FTLE computation. The flow domain (the entire recorded area) was divided into a grid of points where for each grid point, the trajectories of nearby particles were analyzed over a finite time horizon to determine how quickly they diverged or converged. The FTLE value at a point quantifies the rate of separation of particles in the flow, highlighting regions of high mixing (e.g., vortices) or transport barriers (e.g., ridges).

For each replicate, track data were first exported from IMARIS and rescaled. These data were then summarized into a $30 \times 30$ grid vector field using the vfkm tool, applying a smoothing parameter of 0.5. The resulting vector fields were visualized as stream plots to depict the flow dynamics. To identify Lagrangian Coherent Structures (LCS) - surfaces that act as dynamic boundaries between regions of distinct flow behavior - a Finite-Time Lyapunov Exponent (FTLE) field was computed from the vector field. LCS represent the underlying structure of particle trajectories and are revealed as ridges in the FTLE field. The FTLE computation involved two main steps. First, every grid point was used as a seed for trajectory integration across the vector field. This integration was performed using bilinear interpolation in combination with the Runge-Kutta method, over 20,000 time steps with a temporal resolution of $\Delta t = 5$ frames. Notably, the number of integration steps exceeded the number of experimental frames to ensure full coverage of the vector field. Second, the resulting trajectories were used to calculate the FTLE value at each grid point. These scalar FTLE values indicate the degree of divergence in the flow: low FTLE values correspond to regions of low divergence or convergence, while high values mark areas of strong divergence. Ridges in the FTLE map correspond to potential coherent structures that partition the flow into dynamically distinct zones. To further capture the formation of coherent structures, possibly influenced by biological agents such as slime or by environmental currents, a principal tree was fitted to the set of tracked points at each experimental frame. This was done using the SimplePPT algorithm, with parameters $\lambda = 10$ and $\sigma = 10$, and the number of nodes set equal to the number of tracked points at each frame. All methods described above have been implemented and consolidated into a single, reproducible Python package available at: https://github.com/LouisFaure/dyntrack.

### Fixation, sectioning and immunostaining

All specimens were fixed in 4% PFA (in $1\times$ PBS) at 4 °C for a minimum of 6 h, maximum 12 h. Samples of Hexacorallia species were additionally decalcified in EDTA for a period of two weeks[52]. After fixation, all samples were washed with PBS and placed in 15% sucrose for 30 min, 30% sucrose overnight at 4 °C, and then embedded in molds with O.C.T. and frozen at −20 °C. Sectioning was performed using a cryostat set to −20 °C to obtain 20-µm-thick sections, which were collected onto Superfrost slides.

Tissue sections on slides were submerged in Target Retrieval Solution DAKO for antigen retrieval and steamed for 20 min. The solution was then left to cool down for 30–40 min. Sections were washed in PBS containing 0.1% Tween-20 (PBST). A hydrophobic barrier was created with Super PAP Pen surrounding the tissue. The sections were incubated in a humidified chamber overnight at room temperature (22˚C) with primary antibodies (mouse anti-acetylated tubulin ab24610, diluted 1:200, Abcam) diluted in blocking solution ($1\times$ PBS, 3% normal donkey serum, 0.1% Triton-X100). The next day, sections were washed in PBST and incubated with secondary antibodies diluted in PBST for 90 min. Afterwards, slides were washed with PBST, excessive liquid was removed and the slides were mounted using Mowiol as mounting medium. Primary antibodies were detected with following secondary antibodies (Alexa Fluor 647 donkey anti-mouse (1:1000, A31571, Fisher Scientific)). Controls were routinely performed with only the secondary antibody, and, with similar settings as in experiments with primary antibody, no signal was observed.

The specimens were observed with a Nikon Eclipse Ti2-E inverted microscope fitted with Yokogawa W1 spinning disc and Nikon Plan Apo 40×/0.95 /0.11–0.23 WD 0.25–0.17 objective. The sections were excited by the light of following wavelengths: 405 nm, 488 nm, 561 nm, 638 nm. Images were obtained with a Prime BSI camera (Teledyne Photometrics) and processed with NIS Elements and FIJI (ImageJ).

### Statistics and reproducibility

At least three individuals or more were examined per species with few exceptions: for Millepora alcicornis, Agaricia lamarcki, and Gorgonia ventalina, only two colonies were available; Cerianthus sp. was represented by a single specimen. For the recordings of Scyphozoan strobilae, we used two Stomolophus meleagris and two Aurelia coerulea strobilae. Sample size was usually determined for each species by the availability of alive animals. Often, the same individuals were recorded or observed multiple times to ensure the consistency of data prior to the analysis stage. No samples were excluded.

### Reporting summary

Further information on research design is available in the Nature Portfolio Reporting Summary linked to this article.

### Data availability

The datasets generated during and/or analyzed during the current study (i.e. original videos and IMARIS files) are available in the https://doi.org/10.7910/DVN/IMXADV[53].

### Code availability

The computational analysis was performed using a custom pipeline. The source code, publicly available on GitHub, originated from a repository by Bouderlique et al.[10] (https://github.com/LouisFaure/coral_paper) and was subsequently modified for the purposes of this study. The adapted code used to produce all results and figures is available at https://github.com/TBouderlique/cnidarians (https://doi.org/10.6084/m9.figshare.31224049)[54].

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

## Acknowledgements

We thank biological illustrators Dr. Olga Kharchenko and Jessica Eggers for the help with designing figures. I.A. was supported by the ERC Synergy grant (KILL-OR-DIFFERENTIATE), Swedish Research Council, Knut and Alice Wallenberg Foundation, Austrian Science Fund (SFB F78 Program, Emerging Fields "Brain Resilience" consortium, and several Stand-Alone Project Grants). We also thank Natural History Museum of Vienna, Pula Aquarium (Pula), Haus-des-Meeres (HdM) - a public marine aquarium

(Vienna), Research Station CARMABI (Curaçao, Southern Caribbean), and AWI Marine Station Helgoland for cooperation.

## Author contributions
Investigation, data acquisition, writing–original draft: T.K., T.B., K.A., A.F., O.L., A.K., M.M., K.M., J.B., P.F., D.A.-N., I.A. Methodology: T.K., T.B., K.A., A.F., D.A.-N. Conceptualization: O.L., A.K., M.M., K.M., J.B., P.F., D.A.-N., I.A. Validation: K.A., A.F. Resources: O.L., A.K., M.M., K.M., P.F., D.A.-N., Data analysis: T.K., T.B., K.A., A.F., J.B., I.A.

## Funding

## Competing interests
The authors declare no competing interests.
