## [Transparent Peer Review file · Communications Biology]

Cilia-driven surface currents characterize specific cnidarian groups and lifecycle stages

Corresponding Author: Professor Igor Adameyko

Version 0:

Reviewer comments:

Reviewer #1

(Remarks to the Author)

In this manuscript the authors describe the presence of surface ciliary currents in different cnidarian groups and life stages. They further advance the study by clustering taxa according to current organization and interpreting these findings in an evolutionary framework. This is an interesting and timely contribution, highlighting the importance of surface-associated flows in shaping ecological adaptation and evolutionary diversification across cnidarians. However before being considered for publication the authors need to address/rebut some important concerns.

Key topics

The results are frequently intertwined with discussion-like arguments. These sections should either be clearly separated or fully merged. Statements such as “This suggests,” “This could mean,” “We conclude,” or “potentially influenced/involved” should be moved to the discussion.

The discussion section should also be reduced in length. Although the section is introduced at line 343, the authors only begin to effectively discuss their results at line 373. Much of the preceding text would be more appropriately placed in the introduction or methods.

Detailed comments by line

• Results

- o Lines 132–149: The classification of cnidarians should be presented in the introduction, not in the results.
- o Line 179: This belongs to the discussion.
- o Line 192: Conclusions should not appear in the results section.
- o Line 195: This text is discussion material.
- o Line 199: Should be moved to the discussion.
- o Line 206: Discussion content.
- o Line 213: This is a discussion point.
- o Line 230: This does not present results from this study.
- o Line 272: How were the different movement patterns correlated with functional strategies? Please provide supporting evidence.
- o Lines 234–276: This section extensively describes methods and should be moved to that section accordingly. A concise summary could remain here to give readers context before presenting the results.
- o Line 289: Discussion argument.
- o Line 297: This is a discussion point. Please move it to the discussion section and support it with references.
- o Line 312: The relationship between size and the results presented requires supporting evidence. What is the difference in the size range? Where can the comparison be found?
- o Lines 319–324: Methodological description—please move to the methods section.
- o Lines 329–332: Discussion argument.
- o Lines 338–340: Discussion arguments, not results.

Discussion

- Line 427: This statement requires supporting evidence. The manuscript does not provide data on different modes of ciliary beating; please add an appropriate reference.
- Line 432: Please provide supporting evidence or a reference.

- Line 451: This statement needs to be referenced.
- Line 458: Please provide a reference to support this claim.
- Line 465: A reference is required here.
- Line 466: A reference is required here.
- Line 516: While this statement may be valid, it appears far-fetched in the context of the results presented in this manuscript.

Figures

- Figure 1:
 - o Are the scale bars on the IMARIS image applicable to the picture of the actual organism? If so, please clarify. Note that the scale bar differs between the *Fungia* picture and the IMARIS rendering.
 - o Please add scale bars to the images of *Cerianthus*, *Gorgonia*, and *Pteroides*.
 - o Clarify the units of the time scale.
 - o Add the time scale to the *Zoanthus* image.
 - o Why are some lines and text black while others are grey? Please explain the rationale.
- Figure 2:
 - o Same concern as above: why are some lines and text black while others are grey? Please standardize or explain.
 - o The use of different shades of blue is not an effective way to differentiate groups; please consider a clearer colour scheme.
- Figure 3:
 - o This figure largely reproduces data already shown in Figures 1 and 2. Consider merging these figures to reduce redundancy.
 - o The solitary or colonial nature of the species is not consistent across the groups shown. For example, *Fungia* (within Scleractinia) is not colonial. Likewise, not all Zoantharia and Actinaria are photosymbiotic. Please avoid generalizing these characteristics to the entire group.
 - o Clarify what the arrows are intended to indicate.
- Figure 4
 - o What is the size of the magnified region? Please add scale bar.
 - o The cilia density, arrangement, and polarity shown in this image—are they supported by actual data, or are they assumed? If they are assumed, this figure may be misleading, as it visually links presumed polarity, arrangement, and density to flow direction. Please clarify the basis of these features.
 - o Since the white icons also apply to the magnified panel, please remove them from that panel to avoid redundancy.
- Figure 5
 - o Please add scale bars to the animal pictures.
 - o What are the units of the colour code?
 - o What do the shapes represent in the UMAP_1-UMAP_2 graphs represent? and why are they constant across all traits shown?
 - o Do the different colours correspond to species? There is no actual need to add different colours to different species as they are all in different panels. Additionally some of the chosen colours do not contrast well with the light grey making it harder to see the dots.
- Figure 6
 - o The four group numbers should be indicated on the graph.
 - o The heatmap scale for *Zoanthus* appears too small and may be unreadable. Is it different from the scale shown for *Fungia*? Please clarify.
 - o Add scale bars to all organism images.
 - o For the color bar next to *Fungia*: what are the units, and does this scale apply to all other heatmaps (except *Zoanthus*)? Please specify.
 - o For the “liquid’s velocity and hydrodynamic separation” subgraph: what does it contribute to the results already shown in the graph? The figure legend does not explain its relevance—please clarify.

Extended Figures

- Extended Data Figure 1: Without a scale bar it is not possible to judge the scale range mentioned; please add one.
- Extended Data Figure 3:
 - o Please add a scale bar to panel a.
 - o What do the yellow dots in the schematic medusa represent? If they correspond to the yellow boxes in the larger figure, please number them for clarity.
 - o What do the yellow boxes in the magnified region images indicate? Please clarify whether they represent further magnified areas of the already magnified regions, and specify their scale.
 - o Indicate clearly where “magnified region 1” and “magnified region 2” are located.
- Extended Data Figure 4: Please indicate where in the medusa this image was taken, as was done for the previous figure.
- Extended Data Figure 5: As above, please specify the location in the medusa from which this image was taken.

Method details

- Line 657: Were these animals collected from the reef? If so, please provide the collection depth and the collection permit details (if applicable).
- Line 664: Does this correspond to the final density applied to the animals? Please clarify.
- Line 681: Keeping organisms in stagnant water under dark conditions for several hours may reduce oxygen availability and alter animal behaviour. How was the buildup of hypoxic conditions controlled or accounted for? Please add details.
- Line 686: Please provide the light intensity of the CoolLED source.
- Line 690: Please specify the recording speed (frames per second acquisition).
- Line 698: Code details should be included as supplementary information.

- Line 714: Please specify the distance between points.
- Line 719: Please provide justification for this chosen number.
- Line 761: Add details or provide a reference for the decalcification protocol used.
- Line 770: Clarify what “RT” stands for (presumably room temperature) and provide the actual value.
- Lines 770–778: Please introduce the primary and secondary antibodies at the point where they are first mentioned.

Reviewer #2

(Remarks to the Author)

In this study, Koch, Boudierlique and collaborators investigated water currents generated at the body surface of a wide diversity of cnidarians. They tracked the displacement of fluorescent beads around polyps and medusae in a selection of species and observed striking differences between cnidarian groups, and even between species within the same group. Using an analysis pipeline developed in a previous study (Boudierlique et al. 2022), they classified surface current patterns into discrete categories for each sampled species. Finally, they correlated these currents with the density of epidermal cilia in the sampled species, using cryosections, immunostaining, and microscopy. From these analyses, the authors concluded that surface-associated ciliary flows are widespread among cnidarians but not present in all species. They propose that such flows are absent in body forms and species with extensive motility. As the presence or absence of surface-associated ciliary currents appears scattered across the cnidarian phylogenetic tree, the authors suggest that these flows are under strong selective pressure and have emerged and disappeared repeatedly during cnidarian evolution. The work is novel, thorough and nicely complements and extends a previous study on scleractinian corals from the same lab (Boudierlique et al. 2022). However, a few aspects of the manuscript should be clarified or modified prior to publication:

1) The phylogenetic tree topology used in Figures 1, 2, and 3, which forms the basis of the evolutionary discussion (Kayal et al. 2013), is outdated. Notably, it depicts the paraphyly of Anthozoa and a sister-group relationship between Cubozoa and Staurozoa, and between Hydrozoa and Scyphozoa—topologies that have been contradicted by more recent studies. I suggest using a more recent phylogenomic tree (such as Kayal et al. 2018, BMC Ecology and Evolution). The representation of the cnidarian tree in these figures is also confusing, as it is currently split across multiple figures without indicating that the rest of the tree is shown elsewhere. Figure 3 is also largely redundant with Figures 1 and 2. Furthermore, some branches are oddly convoluted, for example, the one leading to Staurozoa. Finally, in Figure 3, “photosymbiosis” is depicted as appearing in the common ancestor of Hexacorallia as well as in the common ancestor of Scyphozoa, with a subsequent loss in Coronatae, based on McFadden et al. 2021. However, this paper does not propose such a scenario but rather multiple, convergent acquisitions of symbiosis within Hexacorallia. A common deep origin of photosymbiosis in Scyphozoa is also not supported by Kayal et al. 2018. Please revise Figures 1, 2, and 3 accordingly.

2) The authors report an interesting difference between polyps and medusae in scyphozoan species, with only polyps showing surface-associated ciliary currents. However, it remains unclear at which developmental stage these currents are lost during the transition from polyp to medusa. Did the authors perform fluorescent bead tracking experiments on the strobila or ephyra (juvenile) stages—both known to be covered with cilia? Investigating these stages could provide further support for their hypothesis that pulsation capacity is anti-correlated with cilia-driven surface currents.

Minor comments:

Line 33: “medusa stages of a lifecycle” this is unclear. Please clarify what is meant here.

Line 82: “More recently.” Remove the period; it should be part of a sentence.

Line 775 and Table 1: Please explain the purpose of the anti-GFP antibody.

Figure 3: What do the black arrows indicate? This should be clearly explained in the figure legend.

Reviewer #3

(Remarks to the Author)

The manuscript reports interesting and original studies of a number of cnidarians from diverse phylogenetic groups. Considerable amounts of data are obtained and carefully analyzed. Interesting patterns are revealed, likely with significant relevance to the biology of these early-diverging metazoans. Generally, the authors were very resourceful in obtaining specimens, although many of these taxa are regarded as “unculturable” by some workers. Ciliary movement likely depends on the organisms being in good condition. A few words about the condition of the organisms should probably be included in the introduction so that readers do not attribute the results to laboratory artifacts. (The immunohistochemistry data speak to this issue quite well.) Throughout, no mention is made of the food consumed by the relevant species, of which there may be considerable diversity, e.g., some actinarians might consume macroscopic prey, while some octocorals may simply absorb dissolved organic matter. Are ciliary currents affected by consumption of food? Is there a diurnal cycle?

Specific comments:

Line 163: “(Boudierlique et al., 2022))”: delete extra “)”

Line 236: “scenario”: “protocol” might be a better term here

Line 267: “Fig.5.”: add space

Lines 279-281: “In fact, these two flow regimes reflect the movements of particles along the corallite ridges away from the mouth and also in the opposite direction – towards the periphery (Fig. 1b-c, Extended Data Fig.2).”: confusing: is the mouth on the periphery? Figure 1 does not have a part “c.”

Line 286: “The bead tracks in this species...” Two species are mentioned, so “these species”

Lines 534- Figures: Figures are complicated but appear to be well done.

Line 645: "The experimental set-up involved a standardize specimen handling,...": should be "standardized"
Lines 656-657: "...after they were used for the undergraduate student marine biology course (species identification and biodiversity study)." Generally, the authors were very resourceful in obtaining specimens. Specimens that were previously used for an undergraduate course, however, might be a little worse for wear.

Version 1:

Reviewer comments:

Reviewer #1

(Remarks to the Author)

The authors have commented and solved most of the previous issues raised by the reviewers and the manuscript has improved considerably. I recommend publication in its current state.

Reviewer #2

(Remarks to the Author)

The authors have appropriately addressed my comments, as well as those of the other reviewers. The revised versions of Figures 1, 2, and 3 are much improved. Congratulations to the authors on this insightful study!

Reviewer #3

(Remarks to the Author)

The authors have carefully and thoroughly revised the manuscript in accordance with the comments in three reviews. An interesting and compelling investigation into these early-diverging metazoans emerges. As elaborated in the discussion, such functional data illuminate the biology of these poorly known creatures. I wonder if rapid cell division of the epithelium poses a constraint on ciliation (i.e., these are unikonts and cannot divide and make a cilium at the same time).

Regarding the revised manuscript, my only concerns are the very minor typographic points listed below.

Line 212: "To further find out, why some cnidarians...": suggest deleting comma

Line 227: "Interestingly, the sensory organs of a medusoid stage- rhopalia were selectively...": suggest moving the hyphen to read: "medusoid-stage rhopalia"

Line 276: "On the other hand, polyps of *Carybdea marsupialis*, a of box jellyfish, demonstrate..." Delete "of"

Lines 340-1: "...marine sponges – another groups of sessile animals with filter-feeding lifestyle and numerous repetitive filtering units, is using surface currents..." Change "groups" to "group" and "is" to "are"

Lines 541-3: "Taxonomic classification is visually encoded using colored squares: orange for Anthozoa, light blue for Cubozoa, sky blue for Staurozoa, dark blue for Scyphozoa, and dark green for Hydrozoa." This is confusing or does not match the figure as rendered in the pdf.

Lines 714-5: "The frames of the original recordings were converted to time codes in seconds, please see the Extended Data Table 2." Suggest: "The frames of the original recordings were converted to time codes in seconds (see the Extended Data Table 2)."

A brief digression concerning the term "video": Currently, this term is used to describe either streaming images or video analog. So, its use in this paper is fine and proper. When reading the older literature, the authors might want to keep in mind that "video imaging" began with actual analog video tape (RS 170 in the USA), with digital images captured from the tape. Thankfully, those days have passed!

Reviewers' comments:

Reviewer #1 (Remarks to the Author):

In this manuscript the authors describe the presence of surface ciliary currents in different cnidarian groups and life stages. They further advance the study by clustering taxa according to current organization and interpreting these findings in an evolutionary framework. This is an interesting and timely contribution, highlighting the importance of surface-associated flows in shaping ecological adaptation and evolutionary diversification across cnidarians. However before being consider for publication the authors need to address/rebut some important concerns.

We would like to cordially thank the reviewer for helping us to improve the manuscript and providing the essential guidance. We hope we resolved all comments with practical improvements, and we believe that with all this advice, our manuscript improved during this revision. Below we provide point-by-point response.

Key topics:

The results are frequently intertwined with discussion-like arguments. These sections should either be clearly separated or fully merged. Statements such as "This suggests," "This could mean," "We conclude," or "potentially influenced/involved" should be moved to the discussion.

We fixed the vast majority of such instances, and now the Results and Discussion and separated and compacted where possible.

The discussion section should also be reduced in length.

We thank the reviewer for this reasonable suggestion. We have substantially condensed and revised many paragraphs of the Discussion. However, some material has necessarily migrated from the Results into the Discussion, which still leaves this section relatively long. In addition, we introduced several important clarifications to address the comments of the other reviewers. We hope to avoid further shortening, as our aim is to place the findings within the context of current knowledge in the field and to present a balanced discussion of several possible biological explanations for our main results.

Although the section is introduced at line 343, the authors only begin to effectively discuss their results at line 373. Much of the preceding text would be more appropriately placed in the introduction or methods.

We appreciate this suggestion and have rearranged parts of the text accordingly. In the revised version, most of such discussion points were moved from the Results section to the Discussion, and the Results was shortened, with some content (more specifically, about species phylogeny or computational analysis) being moved to the Introduction and Methods.

Detailed comments by line

Results

o Lines 132–149: The classification of cnidarians should be presented in the introduction, not in the results.

We moved the classification part to the introduction.

o Line 179: This belongs to the discussion.

Moved to the discussion.

o Line 192: Conclusions should not appear in the results section.

Deleted from results.

o Line 195: This text is discussion material.

Moved to the discussion.

o Line 199: Should be moved to the discussion.

Moved to the discussion.

o Line 206: Discussion content.

Moved to the discussion.

o Line 213: This is a discussion point.

Moved to the discussion.

o Line 230: This does not present results from this study.

We thank the reviewer for this suggestion. However, we believe this point remains valuable in the Results section, as it helps explain why the rophalia may have a more ciliated surface while the bell has fewer cilia, which could make the medusa unable to generate surface flows. These details are too fine and attached to this specific results to be moved to a general discussion.

o Line 272: How were the different movement patterns correlated with functional strategies? Please provide supporting evidence.

This is a great question and we thank the reviewer for raising this. Interestingly, we observed the systematic difference in hydrodynamics patterns (meaning surface currents and their structure), which we of course do not currently know how to explain biologically. We had to say that this difference (being of course not random) most likely correlates with some particular features or functional strategies in different cnidarian animals. We moved this section to the Discussion and improved the wording to reflect it better:

“Within each species, UMAP clustering highlighted that while some track types were predominant, there were also rare trajectory types that contributed to the overall dynamics. At the same time, certain species exhibited highly ordered and repetitive movement patterns, while others showed more stochastic, meandering paths (Fig. 6 – compare panels a and b). This possibly reflects differing functional strategies in surface hydrodynamics, such as efficiency in feeding, mucus transport, or self-cleaning. At this point, we cannot provide an exact biological explanation and functional link for different hydrodynamics patterns, as this requires further studies and shall become a future frontier of cnidarian hydrodynamics research. Some species demonstrated unusually high diversity in their track types, suggesting either a greater range of control over ciliary movements or adaptation to variable environmental conditions.

o Lines 234–276: This section extensively describes methods and should be moved to that section accordingly. A concise summary could remain here to give readers context before presenting the results.

We rewrote this part and left only the general ideas without going into details.

o Line 289: Discussion argument.

Moved to the discussion.

o Line 297: This is a discussion point. Please move it to the discussion section and support it with references.

Moved to the discussion, references added.

o Line 312: The relationship between size and the results presented requires supporting evidence. What is the difference in the size range? Where can the comparison be found?

We thank the reviewer for this interesting question, and we provided necessary clarifications in the text:

*“From the UMAP, it is clear the stereotypicity of the flow structure in scyphozoan and cubozoan polyps contrasts with the greater diversity seen in other cnidarian groups, making scyphozoan and cubozoan polyps an interesting case of streamlined fluid dynamics. Hypothetically, this can be additionally explained by the small size of the polypoid stage (0.5-5 mm, see Figure 7), as compared to larger colonial and solitary cnidarian life forms. This is supported by the observation that larger (5-15 mm) species with similar “polypoid” morphology, such as *Exaiptasia diaphana* and *Zoanthus sociatus*, generate more diverse flow patterns.”*

o Lines 319–324: Methodological description—please move to the methods section.

We believe this remains a very brief description of the method we used, and that it can appropriately stay in the Results section next to the corresponding findings, providing the reader with a quick reminder of the type of analysis performed.

o Lines 329–332: Discussion argument.

Moved to the discussion.

o Lines 338–340: Discussion arguments, not results.

Moved to the discussion.

Discussion

• Line 427: This statement requires supporting evidence. The manuscript does not provide data on different modes of ciliary beating; please add an appropriate reference. and Line 432: Please provide supporting evidence or a reference.

For two previous comments: We thank the reviewer for pointing this out. In the text, we now provide additional references indicating that different modes of ciliary beating and degree of coordination exist in cnidarian species.

“Although ciliary activity at the surface is expected to enhance vertical water movement in general, thereby improving gas exchange and metabolite transport (Shapiro et al., 2014), the emergence of directional surface currents depends on the degree of coordination among cilia (Poon et. al, 2023, 2025; Wan & Poon, 2024) with only sufficiently synchronized beating generating sustained and organized flow patterns.”

- Line 451: This statement needs to be referenced.

We added the references and paraphrased the text:

“Alternatively, the presence or absence of self-generated surface currents in highly colonial cnidarians may reflect differences in sensitivity to ambient flow during early development. Many such species settle in shaded microhabitats, for example under fully grown corals or within reef crevices, where background flow is minimal (Hata et al., 2017; Koehl and Hadfield, 2010).”

- Line 458: Please provide a reference to support this claim.

We added the references:

“As these organisms mature and their colonies expand, they may become less dependent on external or self-generated flow for survival, relying instead on their established structural complexity and improved exposure to ambient flow (Lirman, 2000; Re and Duarte, 2025). For example, one of the modes of propagation of scleractinian corals - through fragmented nubbins or detached branches – might be rather independent of ambient flow conditions for effective resettlement, as scleractinians can effectively propagate in lower-flow, shaded microhabitats, where fallen fragments may readily initiate new colonies (Lirman, 2000).”

- Line 465: A reference is required here.

We thank the reviewer for pointing this out. We clarified that this statement is based on our results:

“According to our results presented here, unlike reef-building scleractinians, octocorals lack the ability to generate surface currents autonomously and instead must rely exclusively on ambient hydrodynamic forces.”

- Line 466: A reference is required here.

We added the reference:

“Also, octocorals are not fragile as compared to scleractinians and do not propagate through fragmented nubbins falling immediately next to the maternal colony (Page et al., 2018).”

- Line 516: While this statement may be valid, it appears far-fetched in the context of the results presented in this manuscript.

We fully agree with the reviewer and we decided to remove this far-fetched statement and leave only this shortened part:

“Although directional coordination of ciliary beating is a fundamental property of many tissues, the underlying regulatory pathways remain poorly understood, not only in cnidarians but even in vertebrates, including humans (Ringers et al., 2023; Wan and Poon, 2024b).”

Figures

- Figure 1:

o Are the scale bars on the IMARIS image applicable to the picture of the actual organism? If so, please clarify. Note that the scale bar differs between the Fungia picture and the IMARIS rendering.

We introduced additional scale bars to have everything correct. Now both images (animal photo and IMARIS) have their own scale bars.

o Please add scale bars to the images of Cerianthus, Gorgonia, and Pteroides.
Of course. This is done.

o Clarify the units of the time scale.

The timescale is represented as a color gradient, showing the beginning and end of the track. Purple indicates the start, while orange/red represents the end. The timescale legend is continual and have seconds in measure. In response to this comment, we added Extended Data Table 2 with conversion of recorded frames to seconds for the list of animals. Every animal creates surface currents with a different speed (so the scale and scale units will differ depending on the recording), and this can be also easily found out by downloading IMARIS files or all actual raw video recordings from Harvard DataVerse (see data accessibility statement). For the general representation of tracks in different animals in main figures we simply show the timeline color code for simplicity. If the readers are interested in specific time or speed of every tracked particle, they will turn to IMARIS files deposited for this study.

o Add the time scale to the Zoanthus image.

Thanks a lot for spotting this! Done, the time scale is added.

o Why are some lines and text black while others are grey? Please explain the rationale.

We made everything the same color. Previously grey color was indicating the groups for which no surface currents were registered. We added some clarifications both on the figure and in the figure description. We hope the reviewer enjoys the improved graphics in the figures, which makes them more accurate and comprehensive. We thank the reviewer for pushing us to do these improvements.

• Figure 2:

o Same concern as above: why are some lines and text black while others are grey? Please standardize or explain.

It is the same logic as above. We now corrected the previous figure design to make it more obvious. We also improved the phylogenetic tree. At this point, we chose grey color to indicate the animal groups, which we did not study (marked n.s.) here – as the animals were not available to us.

o The use of different shades of blue is not an effective way to differentiate groups; please consider a clearer colour scheme.

We fully agree with the reviewer. Now we marked the animal groups differently using the improved colored background squares.

• Figure 3:

o This figure largely reproduces data already shown in Figures 1 and 2. Consider merging these figures to reduce redundancy.

We thank the reviewer for pointing to this. In this revision, we changed figures 1,2,3 the way that they are not redundant. Figure 1 now shows the results of bead tracing only for Anthozoans, and Figure 2 – only for Medusozoa, while Figure 3 is integrating these results on a broader tree to present full evolutionary relationships between the investigated groups.

o The solitary or colonial nature of the species is not consistent across the groups shown. For example, *Fungia* (within Scleractinia) is not colonial. Likewise, not all Zoantharia and Actinaria are photosymbiotic. Please avoid generalizing these characteristics to the entire group.

Yes, we do agree and we removed the pictograms regarding coloniality and other aspects, since they were not directly related to the main idea of the figure.

o Clarify what the arrows are intended to indicate.

We deleted the arrows, they were not necessary indeed.

• Figure 4

o What is the size of the magnified region? Please add scale bar.

Scale bars are added

o The cilia density, arrangement, and polarity shown in this image—are they supported by actual data, or are they assumed? If they are assumed, this figure may be misleading, as it visually links presumed polarity, arrangement, and density to flow direction. Please clarify the basis of these features.

We thank the reviewer for this question and would like to clarify the use of this schematic: firstly, the cilia density is supported by the actual data from the immunohistochemistry shown in this figure (of course, since this is a scheme, everything is schematized). To avoid any speculations beyond the real data, we removed the curved arrows representing individual trajectories. In general, in this figure, we kept only the major arrows showing a general direction of the surface-associated flow (in polyp stage, strobila and ephyra), which is again supported by the data (please also see new supplementary video 2 on strobilae and ephyrae). We thank the reviewer for helping us to make this part of the story and the visual representations more accurate.

o Since the white icons also apply to the magnified panel, please remove them from that panel to avoid redundancy.

We agree. Removed

• Figure 5

o Please add scale bars to the animal pictures.

Scale bars are added

o What are the units of the colour code?

The color bar represents relative values, which are normalized: maximal values correspond to 1, minimal values correspond to 0. These values reflect the different parameters used to study the individual tracks. As they had to be normalized to be compared between species, they represent relative values after such normalization and global comparison procedure. Please see the methods part and the code, and our previous manuscript, where we pioneered this analysis for more details: (Bouderlique et al., 2022).

o What do the shapes represent in the UMAP_1-UMAP_2 graphs represent? and why are they constant across all traits shown?

We thank the reviewer for this question. These shapes represent clusters of data points, and every point is an individual bead track recorded for some studied cnidarian species. The shape of clusters is defined by 2D projection of a multidimensional manifold approximating multiple properties of

numerous bead tracks. Such projections are achieved via a special UMAP non-linear transformation preserving similarity distances in 2D plots from the original full multidimensional analysis space. This is a common practice and well-adopted approach, which we speak about in the improved Methods section. Just to clarify: data points cluster together when they are more similar to one another based on multiple analyzed features of the recorded tracks. The shapes themselves are consistent and permanent for all contributing data points, but the coloring of the dots (panel a) differs in each plot, as it indicates the distribution of values for one specific track feature in the entire dataset (e.g., turns, sinuosity, intersections, speed, length, etc.).

o Do the different colours correspond to species? There is no actual need to add different colours to different species as they are all in different panels. Additionally some of the chosen colours do not contrast well with the light grey making it harder to see the dots.

We changed the colors of highlighted dots to black for all species for better contrast.

•Figure 6
(now Figure 7)

o The four group numbers should be indicated on the graph.

The group numbers are now indicated in the figure and are marked with different color panels.

o The heatmap scale for Zoanthus appears too small and may be unreadable. Is it different from the scale shown for Fungia? Please clarify.

We improved the figure, and now the heatmap scale bar in the left bottom corner is applicable for all FTLE images shown. We removed the redundant heatmap scale bars, because they all were identical and scale to the same values.

o Add scale bars to all organism images.

We added the scale bars to all the photos of cnidarian species.

o For the color bar next to Fungia: what are the units, and does this scale apply to all other heatmaps (except Zoanthus)? Please specify.

The units are arbitrary FTLE values of flow separation and the scale bars are identical for all FTLE images. We improved the figure to make sure this is evident. This is exactly how we performed the analysis in our previous manuscript: (Bouderlique et al., 2022).

o For the "liquid's velocity and hydrodynamic separation" subgraph: what does it contribute to the results already shown in the graph? The figure legend does not explain its relevance—please clarify.

We decided to keep this sub-graph, since it is valuable for understanding what the FTLE images in this figure are showing – i.e. the separation of flow and the hydrodynamic borders between convective cells. We asked several researchers (with different expertise) around, and they think it is rather pedagogically useful. After some doubt, we kept it. We also changed the title for this subgraph to make it clear that it is just the conceptual explanation of how the hydrodynamic separation of a flow looks like on the FTLE images. We also improved the relevant figure in general.

Extended Figures

- Extended Data Figure 1: Without a scale bar it is not possible to judge the scale range mentioned; please add one.

Scale bars are added

- Extended Data Figure 3:
 - o Please add a scale bar to panel a.

Scale bar is added

- o What do the yellow dots in the schematic medusa represent? If they correspond to the yellow boxes in the larger figure, please number them for clarity.

Dots represent the locations of the magnified images on the schematic medusa. We now labeled them with the corresponding letters.

- o What do the yellow boxes in the magnified region images indicate? Please clarify whether they represent further magnified areas of the already magnified regions, and specify their scale.

Yes, indeed, they represent sub-areas of the already magnified regions. They are necessary here to show the cilia density.

- o Indicate clearly where "magnified region 1" and "magnified region 2" are located.

Done

- Extended Data Figure 4: Please indicate where in the medusa this image was taken, as was done for the previous figure.

Done (Extended Data Figure 4,a)

- Extended Data Figure 5: As above, please specify the location in the medusa from which this image was taken.

Done (Extended Data Figure 4, b)

Method details

- Line 657: Were these animals collected from the reef? If so, please provide the collection depth and the collection permit details (if applicable).

Of course. We provided the following information in the Methods:

*“The remaining three species (*Millepora alcicornis*, *Stylaster roseus*, and *Stichopathes* sp.) were examined during a two-week research excursion at the CARMABI research station in Curaçao, Southern Caribbean Sea, in September 2022 (the animals were previously collected from 10 meters depth and hosted in local tanks). The specimens are part of an exchange between the CARMABI Institute (CITES #AN001) and the Natural History Museum Vienna (Naturhistorisches Museum Wien, CITES #AT017). All material was collected in Curaçao (former Netherlands Antilles) under collection permits issued to CARMABI by the Government of Curaçao (permit CARMABI 2019/021824).”*

- Line 664: Does this correspond to the final density applied to the animals? Please clarify.

We thank the reviewer for this comment. In this sentence, the word “density” means the density of the tracking beads material, and we clarified that different density corresponds to the beads of different diameter.

- Line 681: Keeping organisms in stagnant water under dark conditions for several hours may reduce oxygen availability and alter animal behaviour. How was the buildup of hypoxic conditions controlled or accounted for? Please add details.

This is a very important question, and we thank the reviewer for noticing it. During the recordings, water flow was provided to avoid stagnation and to allow proper oxygenation. We have revised the text accordingly:

“Each specimen was covered with 1–2 cm of filtered seawater and allowed to acclimate for two to four hours in an artificial light-free environment with provided water current and conditions similar to maintenance environment.”

- Line 686: Please provide the light intensity of the CoolLED source.

We measured the light intensity using photometer and added the values to the text:

Recordings were performed under a fluorescent stereomicroscope (Leica M165 FC) equipped with a CoolLED light source (light intensity $200\mu\text{mol}/\text{m}^2/\text{s}$ (photosynthetic photon flux)) and a Samsung X23 camera (24 frames per second).

- Line 690: Please specify the recording speed (frames per second acquisition).

We added the recording speed to the text: We measured the light intensity using photometer and added the values to the text:

Recordings were performed under a fluorescent stereomicroscope (Leica M165 FC) equipped with a CoolLED light source (light intensity $200\mu\text{mol}/\text{m}^2/\text{s}$ (photosynthetic photon flux)) and a Samsung X23 camera (24 frames per second).

- Line 698: Code details should be included as supplementary information.

We agree and we removed the code from this section, since it was just indicating that we used ffmpeg to extract every second frame.

“Briefly, frame extraction was performed using ffmpeg, saving every second frame as a TIFF image.” All code and raw data were deposited for downloading according to the journal standards to GitHub and open Harvard Dataverse depository (please see data availability statement).

- Line 714: Please specify the distance between points.

We added the explanation to the text:

“The window size depends on the scale of the animal and is selected so that each particle is tracked every second frame. Therefore, the distance varies according to the video’s frame rate and the specimen’s size.”

- Line 719: Please provide justification for this chosen number.

We added the explanation and the reference to the text:

“Track turns, we estimated by identifying segments where the sinuosity exceeded a threshold of 1.2 (the value at which random sinuosity stops exceeding the observed sinuosity (Limaye et al., 2021), with each such segment counted as a turn.”

- Line 761: Add details or provide a reference for the decalcification protocol used.

The reference for decalcification protocol is added: Samples of Hexacorallia species were additionally decalcified in EDTA for a period of two weeks (Marchioro et al., 2024).

- Line 770: Clarify what "RT" stands for (presumably room temperature) and provide the actual value.

It is indeed the room temperature: "The sections were incubated in a humidified chamber overnight at room temperature (22°C)."

- Lines 770–778: Please introduce the primary and secondary antibodies at the point where they are first mentioned.

Corrected.

.....
We would like to thank the reviewer one more time for helping us to streamline the manuscript and improve the clarity of the text and the visuals. We also performed few new experiments recording strobilae and juvenile scyphozoan ephirae, and we hope the reviewer will appreciate this addition to our story (especially Extended Data Video 2).

Reviewer #2 (Remarks to the Author):

In this study, Koch, Boudierlique and collaborators investigated water currents generated at the body surface of a wide diversity of cnidarians. They tracked the displacement of fluorescent beads around polyps and medusae in a selection of species and observed striking differences between cnidarian groups, and even between species within the same group. Using an analysis pipeline developed in a previous study (Boudierlique et al. 2022), they classified surface current patterns into discrete categories for each sampled species. Finally, they correlated these currents with the density of epidermal cilia in the sampled species, using cryosections, immunostaining, and microscopy. From these analyses, the authors concluded that surface-associated ciliary flows are widespread among cnidarians but not present in all species. They propose that such flows are absent in body forms and species with extensive motility. As the presence or absence of surface-associated ciliary currents appears scattered across the cnidarian phylogenetic tree, the authors suggest that these flows are under strong selective pressure and have emerged and disappeared repeatedly during cnidarian evolution. The work is novel, thorough and nicely complements and extends a previous study on scleractinian corals from the same lab (Boudierlique et al. 2022).

We thank the reviewer for interesting and inspiring comments, and additional guidance. We hope we resolved all comments via performing new experiments and editing the text, where it was requested or advised.

However, a few aspects of the manuscript should be clarified or modified prior to publication:

Below we provide point-by-point response.

- 1) The phylogenetic tree topology used in Figures 1, 2, and 3, which forms the basis of the evolutionary

discussion (Kayal et al. 2013), is outdated. Notably, it depicts the paraphyly of Anthozoa and a sister-group relationship between Cubozoa and Staurozoa, and between Hydrozoa and Scyphozoa—topologies that have been contradicted by more recent studies. I suggest using a more recent phylogenomic tree (such as Kayal et al. 2018, BMC Ecology and Evolution). The representation of the cnidarian tree in these figures is also confusing, as it is currently split across multiple figures without indicating that the rest of the tree is shown elsewhere.

We thank the reviewer for this precious advice, and we followed it to update the phylogenetic tree using more modern sources, including Kayal et al. 2018, BMC Ecology and Evolution and McFadden et al. 2021.

Figure 3 is also largely redundant with Figures 1 and 2.

To some extent we agree, and yet we would like to keep it for the pedagogical purpose and for comparative overview of animals producing and not-producing currents (specifically included in this figure), also showing the position of all investigated specimens across the cnidarian phylogeny. Importantly, we provided a more complete synthesis of results in this exact figure by showing larger groups, which did not make the surface currents – therefore, we reduced the redundancy of figures. We hope the reviewer does not see much problem with keeping it after improvements during this revision.

Furthermore, some branches are oddly convoluted, for example, the one leading to Staurozoa. Finally, in Figure 3, "photosymbiosis" is depicted as appearing in the common ancestor of Hexacorallia as well as in the common ancestor of Scyphozoa, with a subsequent loss in Coronatae, based on McFadden et al. 2021. However, this paper does not propose such a scenario but rather multiple, convergent acquisitions of symbiosis within Hexacorallia. A common deep origin of photosymbiosis in Scyphozoa is also not supported by Kayal et al. 2018. Please revise Figures 1, 2, and 3 accordingly.

We truly appreciate the reviewer's valuable feedback, and we agree. In response, we have revised the trees as suggested to improve the accuracy of data presentation. We changed figures 1, 2, 3 the way that only Figure 3 represents the groups with both detected and not detected currents. Figure 1 shows the results of bead tracing only for Anthozoan and figure 2 – only for Medusozoa, while Figure 3 is summarizing these results and showing them on a broader tree. We removed the pictograms related to photosymbionts and coloniality since they were not directly related to the main idea of the figure. We thank the reviewer for helping us to improve on that.

2) The authors report an interesting difference between polyps and medusae in scyphozoan species, with only polyps showing surface-associated ciliary currents. However, it remains unclear at which developmental stage these currents are lost during the transition from polyp to medusa. Did the authors perform fluorescent bead tracking experiments on the strobila or ephyra (juvenile) stages—both known to be covered with cilia? Investigating these stages could provide further support for their hypothesis that pulsation capacity is anti-correlated with cilia-driven surface currents.

*We thank the reviewer for the helpful suggestion. We performed the additional requested recordings of strobilae and juvenile (freshly-separated) ephyra stages: see Extended Data Fig.3, and Extended Data Vid.2. Interestingly, strobilae of two scyphozoan species (*Stomolophus meleagris* and *Aurelia coerulea*) displayed cilia-mediated directional surface currents flowing from the bottom (foot) to the top with forming ephyrae disks (Fig.3 and Fig.7 also show this with schematic illustrations). The ephyrae also exhibited such flows, as they recently separated from flow-making strobilae, although they were less pronounced than in the polyp and strobila stages. Therefore, the oriented surface-*

associated currents are lost during maturation of ephyrae into medusa. We hope the reviewer specifically enjoys the new video recordings (Extended Data Vid. 2).

Minor comments:

Line 33: "medusa stages of a lifecycle" this is unclear. Please clarify what is meant here.

"Moreover, the presence of surface flow appeared stage-dependent: absent in the medusoid stage but present in the polyp stage of the same species."

Line 82: "More recently." Remove the period; it should be part of a sentence.

Done

Line 775 and Table 1: Please explain the purpose of the anti-GFP antibody.

We thank the reviewer for this question. We originally used anti-GFP antibodies to stain GFP in the coral species. However, because we do not show the GFP channel in the final images for Figure 3 (in order to avoid overloading the figure with unnecessary information, as GFP was not particularly important for our study or for the purpose of the figure) we decided to remove the mention of these antibodies.

Figure 3: What do the black arrows indicate? This should be clearly explained in the figure legend.

We removed the black arrows during this revision, as they were redundant.

.....
Finally, we would like to thank the reviewer one more time for all the additional guidance and for advising to perform particle tracking in Scyphozoan strobilae and ephyrae for better understanding when the surface-associated directional currents are lots in a medusa stage. These experiments made our story more interesting and complete.

Reviewer #3 (Remarks to the Author):

The manuscript reports interesting and original studies of a number of cnidarians from diverse phylogenetic groups. Considerable amounts of data are obtained and carefully analyzed. Interesting patterns are revealed, likely with significant relevance to the biology of these early-diverging metazoans. Generally, the authors were very resourceful in obtaining specimens, although many of these taxa are regarded as "unculturable" by some workers. Ciliary movement likely depends on the organisms being in good condition.

We thank the reviewer for the positive take on our study. This work took several years and many collaborations, which were essential for making sure that the animals are recorded in good conditions. The vast majority of the recorded samples went back to the original aquaria tanks at Hous des Meeres in Vienna (public aquarium), where they continued to live and grow. Our recording conditions were similar to these where the animals were raised initially in captivity, which very few exceptions (such as sea pens, which we had to dig out from the original aquaria substrate at Aquarium Pula and re-arrange for recordings with the use of a stereo-microscope; or few species recorded at Halgoland and Curacao – a similar situation). As a rule of a thumb, for imaging, the individuals were temporarily transferred to a smaller vessel positioned under the microscope. Prior to recording, the specimens were allowed to acclimate and resume normal behavior without being disturbed during 1-4 hour with ventilation, including polyp extension, to ensure that observations reflected natural conditions. We improved the

methods sections and provided the additional information (including light and water conditions, also see Extended Data Table 1 for feeding) about upkeep of the animals.

A few words about the condition of the organisms should probably be included in the introduction so that readers do not attribute the results to laboratory artifacts. (The immunohistochemistry data speak to this issue quite well.)

As suggested by the reviewer, we added a notion about the upkeep of animals in the beginning of our Results:

“Of note, the majority of specimens examined in this study were successfully reared and maintained in public aquaria (Haus des Meeres in Vienna and Pula Aquarium in Pula). Imaging was conducted within short periods of time and under conditions similar to routine husbandry parameters, in order to preserve animal health and maintain their native surface hydrodynamics.”

Throughout, no mention is made of the food consumed by the relevant species, of which there may be considerable diversity, e.g., some actinarians might consume macroscopic prey, while some octocorals may simply absorb dissolved organic matter. Are ciliary currents affected by consumption of food? Is there a diurnal cycle?

We thank the reviewer for this question. As the vast majority of samples was obtained from the public aquaria cultivating these animals routinely (Haus des Meeres and Vienna and Pula Aquarium, Croatia), we did not get into specific details of the upkeep of every species (as we recorded a wide diversity of them). We simply relied on the fact the animals successfully grew, and even propagated - in some cases.

Therefore, we did not specifically investigate whether differences in food type or feeding behavior might contribute to the presence, absence, or specific features of the surface flows. This shall be a matter of a future study, because it does increase the numbers of recordings, formation of experimental and control groups for multiple species. This would be impossible to achieve within this overview study, which primarily focuses on the cnidarian biodiversity.

Also, because of the similar reasons, we did not assess potential diurnal variations. In part, we did not systematically look into diurnal variation because we already performed the preliminary experiments using night and day recordings for several octocorals and scleractinian corals. These preliminary experimentations did not reveal any specific difference, which cannot be explained by hidden or extended polyps during day or night phase. Because we could not continue these experiments systematically for all reported species, we decided to make it a topic of future studies.

To accommodate this comment more practically, we introduced a statement to the manuscript to reflect that we did not investigate these aspects, and they shall become the questions of future studies: “Important open questions also concern the physiological control of these surface currents - both nervous system-dependent and independent - as well as their modulation by diurnal cycles and feeding dynamics, all of which lie beyond the scope of this study. A further critical challenge is to elucidate the developmental and molecular mechanisms that establish collective ciliary orientation across polarized epithelial fields (Wallingford, 2010).”

Also, we added the specific passage to the Methods section:

“With the exceptions described below, all animals were reared, maintained, and propagated ex situ at Haus-des-Meeres (HdM), a public marine aquarium in Vienna, Austria. Specimens were kept in closed-circuit seawater systems under stable environmental conditions and standardized husbandry protocols, and many colonies were sustained over prolonged periods with regular growth and

propagation. Routine monitoring of behaviour, polyp extension, and tissue appearance indicated that the animals remained in good health throughout the study. Details of feeding regimes and water temperature settings for each species are provided in Extended Data Table 1."

As the reviewer can see, during this revision we also added Extended Data Table 1 listing species with corresponding husbandry conditions including the type of feeding and temperature.

Specific comments:

Line 163: "(Bouderlique et al., 2022)": delete extra ")"

Done.

Line 236: "scenario": "protocol" might be a better term here

Thank you! Done.

Line 267: "Fig.5)": add space

Done.

Lines 279-281: "In fact, these two flow regimes reflect the movements of particles along the corallite ridges away from the mouth and also in the opposite direction – towards the periphery (Fig. 1b-c, Extended Data Fig.2)": confusing: is the mouth on the periphery? Figure 1 does not have a part "c."

Thank you! It does not have it indeed, we corrected the text:

"In fact, these two flow regimes reflect the movements of particles along the corallite ridges towards the mouth and also in the opposite direction – away from the mouth (Fig. 1b, Extended Data Fig.2)."

Line 286: "The bead tracks in this species..." Two species are mentioned, so "these species"

Thank you for noticing! Corrected.

Lines 534- Figures: Figures are complicated but appear to be well done.

Thank you! We improved the figures to make them more clear by removing redundant elements and adding colored blocks.

Line 645: "The experimental set-up involved a standardize specimen handling,...": should be "standardized"

Corrected.

Lines 656-657: ",,after they were used for the undergraduate student marine biology course (species identification and biodiversity study)." Generally, the authors were very resourceful in obtaining specimens. Specimens that were previously used for an undergraduate course, however, might be a little worse for wear.

The specimens were briefly recorded for bead-tracking experiments and only afterwards were given to students for their course work (exactly to preserve their good conditions). Throughout this study, we were able to access this wide range of species only because of our collaborations with Haus des Meeres (public aquarium of Vienna), Pula Aquarium (Pula, Croatia), University of Vienna and Museum of Natural History in Vienna. Also, our colleagues from Curacao (CARMABI), Helgoland and other marine

biological stations were very supportive. And, of course, this project took several years of work to accumulate the recordings from multiple species.

.....

We thank the reviewer for helping us to improve how we describe the animal conditions and their upkeep, and for providing other useful advice. We hope the reviewer will be happy with the revised manuscript.

REVIEWERS' COMMENTS:

Reviewer #1 (Remarks to the Author):

The authors have commented and solved most of the previous issues raised by the reviewers and the manuscript has improved considerably. I recommend publication in its current state.

Reviewer #2 (Remarks to the Author):

The authors have appropriately addressed my comments, as well as those of the other reviewers. The revised versions of Figures 1, 2, and 3 are much improved. Congratulations to the authors on this insightful study!

Reviewer #3 (Remarks to the Author):

The authors have carefully and thoroughly revised the manuscript in accordance with the comments in three reviews. An interesting and compelling investigation into these early-diverging metazoans emerges. As elaborated in the discussion, such functional data illuminate the biology of these poorly known creatures. **I wonder if rapid cell division of the epithelium poses a constraint on ciliation** (i.e., these are unikonts and cannot divide and make a cilium at the same time). Regarding the revised manuscript, my only concerns are the very minor typographic points listed below.

We thank the reviewer for raising this interesting point regarding a potential constraint linking cell division and ciliation.

Indeed, the ciliary apparatus is typically resorbed prior to mitosis, with the basal body repurposed for spindle formation, and then reassembled after cell division is complete.

While our current study did not directly investigate cell cycle dynamics, our immunofluorescence data reveal dense ciliation in those species and life stages that are capable of generating structured surface currents. This correlation leads us to hypothesize that the patterned currents themselves may provide a stabilizing spatial cue. The synchronous beating of densely packed, pre-existing cilia establishes a local hydrodynamic and possibly signaling environment. A newly formed daughter cell, integrating into this epithelium, would therefore inherit not only a positional pattern but also the dominant beating orientation from its neighbors. This could ensure that even as individual cells divide and reassemble their cilia, the overall tissue-level ciliary pattern and coordinated beating required for effective current generation is maintained.

We want to point that this specific mechanistic hypothesis on pattern inheritance is out of the focus of our current study and remains a speculative point. As such, we will not include this hypothesis in the manuscript.

Line 212: "To further find out, why some cnidarians...": suggest deleting comma
Thank you! Comma is deleted.

Line 227: "Interestingly, the sensory organs of a medusoid stage- rhopalia were selectively...": suggest moving the hyphen to read: "medusoid-stage rhopalia"
Thank you! We changed it.

Line 276: "On the other hand, polyps of Carybdea marsupialis, a of box jellyfish, demonstrate..."
Delete "of"
We deleted it.

Lines 340-1: "...marine sponges – another groups of sessile animals with filter-feeding lifestyle and numerous repetitive filtering units, is using surface currents..." Change "groups" to "group" and "is" to "are"
We corrected it, thank you!

Lines 541e-3: "Taxonomic classification is visually encoded using colored squares: orange for Anthozoa, light blue for Cubozoa, sky blue for Staurozoa, dark blue for Scyphozoa, and dark green for Hydrozoa." This is confusing or does not match the figure as rendered in the pdf.

Thank you for pointing that out. Indeed, the final part of the figure description did not match the updated version of Figure 3. When making the new version of Figure 3 we removed the color coding to eliminate redundancy and improve visual clarity. The caption has now been corrected to accurately reflect the figure.

Lines 714-5: "The frames of the original recordings were converted to time codes in seconds, please see the Extended Data Table 2." Suggest: "The frames of the original recordings were converted to time codes in seconds (see the Extended Data Table 2)."

We moved the Supplementary Table 2 to the "Data Availability section" where we provided the direct link. We also added the link in the Methods section:

"The frames of the original recordings were converted to time codes in seconds (see here: <https://dataverse.harvard.edu/dataset.xhtml?persistentId=doi:10.7910/DVN/IMXADV>).

To avoid confusions with the numeration: We added Supplementary Data 1 for listing exact numbers of tracks for species/groups (Fig.6a,b). We also moved Table 1 to Supplementary (it is named Supplementary Data 2)

A brief digression concerning the term "video": Currently, this term is used to describe either streaming images or video analog. So, its use in this paper is fine and proper. When reading the older literature, the authors might want to keep in mind that "video imaging" began with actual analog video tape (RS 170 in the USA), with digital images captured from the tape. Thankfully, those days have passed!

Thank you for the clarification! We agree, and we have kept the term "video" as used in the manuscript bearing in mind its historical context in earlier literature. Nevertheless, we chnded "Supplementary Video" to "Supplementary Movie" as it is required in the guidelines.